# A unified model framework for the multi-attribute consistent periodic vehicle routing problem

**Maria Gulnara Baldoquin[1], Jairo A. Martinez[1], Jenny Díaz-Ramírez[2]***

**1** Department of Mathematical Sciences, Universidad EAFIT, Medellin, Antioquia, Colombia, **2** Engineering Department, Universidad de Monterrey, Monterrey, N.L., Mexico

* jenny.diaz@udem.edu

## Abstract

Modeling real-life transportation problems usually require the simultaneous incorporation of different variants of the classical vehicle routing problem (VRP). The periodic VRP (PVRP) is a classical extension in which routes are determined for a planning period of several days and each customer has an associated set of allowable visit schedules. This work proposes a unified model framework for PVRP that consists of multiple attributes or variants not previously addressed simultaneously, such as time-windows, time-dependence, and consistency -which guarantees the visits to customer by the same vehicle-, together with three objective functions that respond to the needs of practical problems. The numerical experimentation is focused on the effects of three factors: frequency, depot centrality, and the objective function on the performance of a general–purpose MILP solver, through the analysis of the achieved relative gaps. Results show higher sensitivity to the objective functions and to the problem sizes.

**Data Availability Statement:** All relevant data are within the manuscript and its Supporting Information files.

**Funding:** MGB: Universidad EAFIT: Grant 881000027. JAM: COLCIENCIAS - Young

## Introduction

Transport decisions in modern companies are made in the context of integrated supply chains. Tactical and operational levels of transport comprise medium and short-term decisions, including detailed planning of visit schedules, routes and load plans. Correct synergy between such decision levels contributes to the consolidation of the supply chain and is a recurrent challenge for planners [1, 2].

The vehicle routing problem (VRP) is a problem that has been widely used for the representation of distribution activities and transportation of goods [3]. The VRP is based on a set of points and available vehicles, and for each vehicle the points to be visited and their order are decided. The classic objective is that the total distance covered by each vehicle is the minimum and that each point receives exactly one visit. Features of particular case studies are added to the basic VRP model. A taxonomy of the VRP model family is detailed in [4]. For the study of supply chain management problems, variants have been created that emphasize tactical aspects such as the periodic vehicle routing problem (PVRP), while others emphasize operational aspects, such as the vehicle routing problem with time windows (VRP-TW) and the time

Researchers and Innovators Program Number 812. The funders had no role in study design, data collection and analysis, decision to publish, or preparation of the manuscript.

**Competing interests:** The authors have declared that no competing interests exist.

dependent vehicle routing problem (TDVRP). Mixed integer linear programming (MILP) models are proposed to represent these variants, in which points of interest are usually called customers and the starting point is called depot [3–5]. To ease the reading, Table 1 lists the meaning of the acronyms used throughout the paper.

In the VRP-TW each customer must be visited within a certain time interval. In TDVRP, the travel time between two customers depends on the state of traffic at the time of departure. The PVRP looks for building a plan of optimal routes for the entire planning horizon (i.e. more than one day), knowing in advance the frequency of visits demanded by each customer. It involves deciding the pattern of visits for each customer, the selection of the vehicles that visit each customer, and the visit order. In the literature we find models for the VRP that simultaneously capture two of these variants: PVRP and VRP-TW or TDVRP and VRP-TW [6, 7]. Three important variants of the PVRP are revised in [8]: the PVRP with time windows (PVRP-TW), the multi-depot PVRP, and the PVRP with service choice, which includes the service frequency as a decision variable, given that customers are visited a given number of times over the period, with a schedule that is chosen out of a menu of schedule options.

Some papers deal with additional constraints that improve customer service quality in VRP problems. In [9] a consistent VRP (ConVRP) considers that the same driver visits the same customers throughout the planning horizon, at roughly the same time on each day that these customers are visited. In [10] a generalized ConVRP is considered, where each customer is visited by a limited number of drivers and the variation in the arrival times is penalized in the objective function. A collection of vehicle routing problems in which consistency considerations are relevant are described in [11]. The consistent PVRP (ConPVRP) problem is referred with this name in [12] but it was previously addressed in [2, 13]. The problem addressed in [2, 13] is a consistent PVRP with time windows (ConPVRP-TW) where the model selects a pattern of visits to each customer, according to its frequency of visit, which depends on the customer's sales volume: weekly (the same day each week; e.g. every Tuesday), semiweekly (two visits per week; e.g. Monday and Thursday or Tuesday and Friday), bimonthly (2 times a month; e.g. in the first and third weeks or in the second and fourth weeks, but always on the same day of the week), and monthly. The objective function considered is the sum of travel time needed to supply all customers. In [2], the problem was divided into phases, and a heuristic solution method is highlighted for Phase 2. A non-linear model was solved with a heuristic method that uses in one of its steps a well-known integer linear problem as a black box. In [13], a model and two heuristics are proposed to solve the ConPVRP just described.

All the previous ones have been useful for the analysis of prototypical cases, nevertheless, there are problems that require to consider simultaneously periodicity, time windows and

**Table 1. List of acronyms.**

| Acronym | Meaning of the acronym | Acronym | Meaning of the acronym |
|---------|------------------------|---------|------------------------|
| SPTW | Shortest Path Problem with Time Windows | ConVRP | Consistent VRP |
| ESPTW | Elementary SPTW | PVRP | Periodic VRP |
| ESPTWQ | ESPTW and capacity constraint | TDVRP | Time Dependent VRP |
| TSPTW | Traveling Salesman Problem with Time Windows | PVRPTW | PVRP with Time Windows |
| CVRP | Capacitated VRP | TDPVRP | Time Dependent PVRP |
| DCVRP | Distance-Constrained VRP | TDPVRP-TW | TDPVRP with Time Windows |
| VRPSDP | VRP with Simultaneous Pickups and Deliveries | ConPVRP | Consistent PVRP |
| VRPB | VRP with Backhauls | ConPVRP-TW | Consistent PVRPTW |
| VRPTW | VRP with Time Windows | ConTDPVRP | Consistent TDPVRP |
| VRPTWPD | VRPTW with Pickups and Deliveries | ConTDPVRP-TW | Consistent TDPVRP-TW |

time dependency, as the design of routes for companies of distribution of goods in urban centers [2]. In this context, using a model that ignores any of the variants can lead to inefficient plans that generate additional costs and high percentages of non-compliance.

This article presents a unified modeling framework for the ConPVRP, where the service pattern for a given frequency is a decision of the model. This modeling framework combines the variants of ConPVRP-TW, consistent and time dependent PVRP (ConTDPVRP), and consistent and time dependent PVRP with time windows (ConTDPVRP-TW); taking into account two different objective functions that are not so usual but important in some real applications: the minimization of the maximum duration of a route, which is related to operating costs, the minimization of the time in which the last customer is visited, which is related to their degree of satisfaction [14], and also an objective function more used in literature: minimization of the total transportation time over the planning horizon [12].

A general real-world context that inspires the development of the framework for ConPVRP is as follows: The company has a set of sale points (or customers) that must be visited at a frequency determined by its sales volume (or demand) by one of the trucks of the company fleet. There are 4 types of visit frequency: weekly (the same day each week, for example every Tuesday), biweekly (two visits per week, e.g. Monday and Thursday or Tuesday and Friday), bimonthly (2 times a month, e.g. in the first and third weeks or the second and fourth weeks, but always on the same day of the week), and monthly. Trucks must start and finish their journey at the central depot. Each customer must always be visited by the same truck. Several customers can be visited on the same day by the same truck. The trucks are available from Monday to Friday. Though the trucks have a limited capacity, it is assumed that the total available travel time is the dominant constraint [2]. This is why $f1$ and $f2$ become relevant objective functions.

This paper makes two contributions to literature, being the first one of modeling-type, and the second one a numerical-type contribution; as follows:

1. A unified model framework for the multi-attribute ConPVRP inspired by real problems. The framework includes variants, and their relationships, not considered simultaneously before. In addition, it includes the analysis of three objective functions, two of them uncommonly discussed but inspired by real problems.

2. The experimentation design includes the simultaneous analysis of three relevant factors to these types of problems: frequency, depot centrality, and the objective functions. We provide experimental evidence of how the two non-conventional objective functions are harder problems to solve, and that some active constraints found in the literature actually do not improve the performance of the model.

The remainder of this paper is structured as follows: Detailed literature review is presented in next section. Then we present the formulation of the modeling framework. Next, we describe the experimental design, the experimental results and their respective analysis. Finally, we summarize our work and propose lines for future research.

## Literature review

This literature review has been divided into three subsections. The first one contains the main proposals for the modeling and solution of the PVRP, including previous work related to ConPVRP. Next subsection explores the research dedicated to TDVRP, emphasizing techniques for formulating time dependent travel times or velocities, which inspired the way the time dependence issue is addressed in the proposed framework. Finally, the last subsection mentions different researches oriented to obtain modeling frameworks for the multi-

attribute VRP. It is worth noting that no previous studies addressing the TDPVRP variant were found. In particular we have not found MILP formulations for the ConTDPVRP and ConTDPVRP-TW cases, both included in the modeling framework proposed in the present work.

## Periodic VRP and variants

The PVRP was presented in a seminal article by Beltrami & Bodin at 1974 in the context of a route design problem for garbage collection [15]. Since its inception, multiple variants have been added to the PVRP combining tactical and operational features for real case analysis. Russell et al. [16] addressed a problem close to PVRP for planning weekly visits and balancing the vehicle requirements. Christofides et al. [17] proposed a formal definition of the PVRP and identified the three decisions that make up the problem: (i) selection of visit patterns, (ii) selection of customers that will be visited by each vehicle during each day, and (iii) definition of the order of visits. They presented an integer programming (IP) formulation, in which the PVRP problem was interpreted as an extension of the routing problem with a pattern selection decision included.

In [8] two approaches to the PVRP are distinguished depending on how the decisions are prioritized: Assignment routing problem if the selection of visit patterns is prioritized, and periodic routing problem when the construction of routes is prioritized. The first approach is applied to cases in which tactical decision prevails, as is the case of [2] and [13], while the second approach is specific to situations in which the interest lies in operational decisions.

There are several precedents of the inclusion of additional operational features in the PVRP. In [6] the PVRP was extended to the PVRP-TW and used a modified data set for the PVRP including randomly generated time windows for each customer. Cantu et al. [18] investigated a multi-depot periodic vehicle routing problem (MDPVRP) with due dates and time windows motivated by the case of a Mexican brewing company. The authors used a set of artificial data constructed from real information provided by the brewing company. Related works adding the constraint of consistency of the drivers along the visits of the planning horizon can be found in [13] and [12]. Table 2 shows a summary of the features dealt in these works.

There are some works aimed at obtaining stronger formulations for PVRP variants. In [12] valid inequalities for a consistent PVRP are derived from the generalized multistar inequalities presented in [19], where vehicles are considered of unitary capacity. In [20] the authors introduced the flexible periodic vehicle routing problem (FPVRP), and derived valid inequalities related to the flow balance. They implemented optimality cuts regarding the load of the

**Table 2. Summary of consistent-VRP review.**

| Paper | Type of model | Attributes | Objective function | Solution method | Number of customers | Dataset |
|---|---|---|---|---|---|---|
| [2] (2014) | Mixed integer non linear non convex program | Scheduling visits to customers located in the same cluster in a given period | Minimizing the total transportation time over the planning horizon | Hybrid heuristic + CPLEX solver | 100-1000 | A. Escalera |
| [13] (2018) | Mixed integer non linear program | ConPVRP-TW, PVRP-DC | | Hybrid heuristic: local search + Gurobi solver | 400-2000 | A. Escalera |
| [12] (2019) | Integer program | PVRP-DC, capacitated | Minimizing the total routing cost | Branch & Cut | 11-71 | Author's own construction |

PVRP: Periodic VRP, ConPVRP: Consistent VRP, TW: Time-windows, DC: Driver Consistent.

vehicles when returning to the depot. Both works incorporated constraints for symmetry breaking based on the indexation of the customers. It is important to note here that in recent literature research on valid inequalities has concentrated mainly on variants of the capacitated VRP, see for example [21].

During the first decade, much of the research on the PVRP followed Beltrami's [15] two phase solution methods. Russell et al. [22] presented constructive heuristics while Christofides [17] introduced relaxations to the problem and a two-phase solution method for them. Tan [23] used a heuristic algorithm based on a previous work of Fisher & Jaikumar [24] for solving an IP formulation. Both works prioritized the selection of visiting patterns. Chao et al. [25] adopted an approach similar to Russell & Igo, using an improvement phase after assigning the visit pattern to each customer.

Cordeau et al. [26] presented a tabu search algorithm applicable to the PVRP-TW. Subsequently Drummond et al. [27] proposed a metaheuristic based on genetic algorithms, where the intensification strategy was reinforced by local search. Regarding exact solution methods, Francis et al. [28] developed procedures on Lagrangian relaxation of the PVRP formulation as a linear programming problem. Mourgaya et al. [29] solved a tactical version of the PVRP using the column generation method. However, to reduce computational complexity, they chose to solve the subproblems using heuristic techniques. Finally, Vidal et al. [30] proposed an algorithmic framework for the multidepot PVRP with capacitated vehicles and constrained route duration, combining population-based evolutionary search and neighborhood-based metaheuristics.

## Time dependent VRP

The TDVRP was first presented in [31] and [32] in 1991 to address VRPs taking into account the effect of vehicle congestion on plan performance, in urban contexts. A state of the art review for the TDVRP variant is presented in [33]. The pioneering work of Malandraki & Daskin [31] and Ahn & Shin [32] raised the need to include in the models the variation in travel times due to traffic congestion and occasional factors such as accidents. Ahn & Shin understood the problem as a natural extension of the VRP-TW, while Malandraki & Daskin posed a situation without time windows. Both works were interested in the computational complexity of the variant: The last one by taking the case of the time dependent travelling salesman problem, while Ahn & Shin investigated the increased complexity of the problem with respect to a problem with time windows and constant travel time, identifying the non-passing property, later referred to as FIFO in [34], as a desirable condition also related with the complexity of the problem.

A MILP-type model for the TDVRP was formulated in [31], and their key contribution was the division of the day into time intervals and the definition of a stepwise travel speed function over such intervals. The idea of dividing the day into time intervals was taken up in the work of several authors such as Ichoua et al. [34] and Figliozzi et al. [7]. However, to ensure the satisfaction of the FIFO property, these authors proposed stepwise speed functions and calculation of travel time by integration. Most of the current formulations are based on these models, highlighting their application to new variants such as the green VRP, which aims to minimize fuel consumption by establishing routes and appropriate schedules [35–37].

The first solution methods for TDVRP were modifications to constructive algorithms for VRP. In [31] authors adapted the nearest node insertion heuristics and the sequential route construction when there is time dependency, to deal with TDVRP. Ahn et al. [32] modified Clarke & Wright savings algorithm to solve a TDVRP-TW. Ichoua et al. [34] used tabu search metaheuristics, and adapted operations to account for time dependency in a TDVRP while

Hashimoto et al. [38] used iterated local search to resolve TDVRP-TW formulation where time windows are soft, and like Ichoua et al. made significant modifications to address time dependence.

In [39] authors investigated the adaptation of algorithms used in the VRP to solve the TDVRP. When they used metaheuristic based on local search, they recognized that the improvement of a feasible solution does not only affect the travel times of the customers involved in the operation, but instead impacts all nodes assigned to a route. A similar phenomenon was reported in [40] and [41] when trying to solve the TDVRP by tabu search, defining specific (2-opt) neighborhood movements to deal with time dependence. Donatti's ant colony optimization metaheuristics proposal and Figliozzi's route improvement algorithms [7] were presented to deal with the TDVRP without relying on standard local search procedures and reporting solutions at least as good as those of their counterparts.

## Multi-attribute VRP and modeling frameworks

In [42] authors coined the term attribute to refer to the variants, characteristics and types of decisions that appear in real VRPs. They identified fifteen VRP multi-attributes (MA) that have been the object of intense study in the literature and the heuristic and metaheuristic techniques used for their solution. Later in [43] the authors described the development of a solver for MA-VRPs, which they called unified hybrid genetic search metaheuristic and evaluated its performance through computational experimentation on different instances considering multiple periods, multiple depots, generalized time windows, time dependence, between other attributes. These authors focused on the approximate methods of solution but didn't present a modeling framework that encompasses the variants that were considered.

In the literature there are several works in which "modeling frameworks" are presented for families of VRP variants. The Table 3 presents a summary of the most relevant works. Desaulniers et al. [44] developed a modeling framework based on integer nonlinear mixed programming. Although the authors include the derivation of the VRP-TW from the proposed framework, they do not account for periodic or time-dependent problems. Irnich [45] present a unified modeling framework whose formulation makes use of a routing graph in which any solution to a MA-VRP is represented by a single cycle called a giant tour. The Irnich's modeling framework includes among other problems the VRP-TW and the PVRP-TW. The framework proposed by Desaulniers [44] is designed for the solution using Branch & Bound (B&B) methods and column generation, while the Irnich's framework [45] is aimed at the efficient implementation of metaheuristics.

Subsequently Puranen et al. [46] present a modeling language or metamodel for the MA-VRP based on formulations on graphs and assignment functions. In this metamodel the

**Table 3. Summary of frameworks for VRP variants.**

| Paper | Type of model framework | Included variants | Objective function | Projected solution method |
|---|---|---|---|---|
| [44] (1998) | Mixed integer non linear programming | ESPTW, SPTW, ESPTWQ, TSPTW, VRPTW, VRPTWPD | Minimizing the total cost of routes | Column generation, Lagrangian relaxation |
| [45] (2008) | Formulation on graphs: routing graph, giant tour representation | CVRP, DCVRP, VRPSDP, VRPB, PVRP, PVRPTW | | Metaheuristics based on local search |
| [46] (2011) | Metamodel based on formulation on graphs | CVRP, VRPTW, VRPSDP, VRPB, PVRP, TDVRP, PVRPTW | | Metaheuristcs based on local search |

SP: Shortest Path Problem, E: Elementary, TW: Time Windows, Q: capacity constraint, TSP: Traveling Salesman Problem, B: Backhauling, P: PVRP, TD: Time Dependent, C: Capacitated, DC: Distance-Constrained VRP, SDP: Simultaneous Pickup and Deliveries.

VRP variants are not formulated as mathematical programming problems, but it is able to express many variants among which VRP-TW, PVRP, PVRP-TW, and TDVRP stand out. Puranen's modeling language is aimed at the efficient implementation of metaheuristics [46]. The consistency attribute is not mentioned in any of the three modeling frameworks mentioned above.

The work of Rodríguez et al. [12] deals with the ConPVRP. However, our proposal presents multiple differences with respect to this one: In [12] the model considers a minimum number of customers per route, without clarifying the real-life problem conditions that justify such an assumption; in contrast, the models in our framework do not necessarily consider a minimum or maximum number of customers per route. We consider two non-conventional objective functions, while in [12] it is considered a more classical objective function. There are also methodological differences: in the experimental phase we evaluate the influence of different characteristics of the constructed instances on the solution, and the maximum computation time set is half of the time dedicated in that work.

During the construction of the state of the art we did not find any reference in which a model is formulated for the ConPVRP that includes the attributes of time windows and time dependency.

## Model framework

The model framework proposed in this work considers the following variants of the ConPVRP: ConPVRP-TW, ConTDPVRP and ConTDPVRP-TW. First, the structure of the model framework is schemed in Table 4. Next, the objective functions selected are described and justified, and finally, the model framework is presented in detail.

Table 4 schematizes the variants that are analyzed and distinguishes the specific constraints of each variant from the core constraints of the ConPVRP, by indicating the numbers of the equations that constitute each one. The core VRP constraints reflecting consistency and periodicity are included in the column "all models". Constraints in the next column are needed in the models that don't consider time-dependence, in contrast with the fourth column that identifies the constraints that are exclusively used if the model considers time-dependence. The time-windows variant requires the addition of the constraints in the last column. The additional valid constraints that were revised in the numerical experimentation are also optional and considered in all models.

The set of constraints shaping all variants considered in this work can be used to optimize the function that better fulfil the researcher's needs. The objective functions equations included in the framework, and the constraints needed to connect them with the rest of the model, are indicated in their respective cell in Table 4.

The model framework proposed includes three options of objective functions:

**Table 4. Summary of model framework.**

| Objective functions | Constraints corresponding to conPVRP models included in the framework | | | |
|---|---|---|---|---|
| | All models | Models without TD | Models with TD | Models with TW |
| | ConPVRP, ConPVRP-TW, ConTDPVRP, ConTDPVRP-TW | ConPVRP, ConPVRP-TW | ConTDPVRP, ConTDPVRP-TW | ConPVRP-TW, ConTDPVRP-TW |
| **For all three functions** | (4–13), (32) + additionals: (28–30), and (31) instead (11) | (17) | (18), (21–25) | (14–16) |
| **Only for $f1$** | (1), (26) | (20) | | |
| **Only for $f2$** | (2), (27) | (19) | | |
| **Only for $f3$** | (3) | | | |

- (*f*1) that minimizes the maximum duration of a route.

- (*f*2) that minimizes the time in which the last customer is visited.

- (*f*3) that minimizes the total transportation time over the entire planning horizon.

(*f*1) and (*f*2) are functions of the makespan minimization type. According to Braekers et al. [4], they are not considered standard objective functions although both are based on time or distance. These functions have been used in parcel applications [47], load balancing in home health services [48], manufacturing processing times [49]. Other practical problems where (*f*2) gains importance is in bus routing, where the maximum travel time of the first student collected in the route wants to be minimized [50, 51]. (*f*3) has been added in the analysis due that it is one of the standard and most common objective functions explored in VRP. This inclusion will allow future benchmark or experimental comparisons. Examples of VRP studies considering this function are [7, 12, 20–22, 39, 52].

## The mathematical model

### Scalar parameters

$n$: number of customers

*t_days*: number of days in the planning horizon

$m$: number of available vehicles

*p_num*: number of visiting frequencies

$D$: maximum working day length

$h$: number of time intervals in a day

$\epsilon$: A positive small enough number

### Indices

$i, j$: customers, depot

$l$: days

$k$: vehicles

$p$: visiting patterns

$u$: intervals that make up a day

$f$: visiting frequencies

### Sets

$V := \{1, \cdots, n + 2\}$: the first element in $V$ is the initial depot, the last element is the "final" depot (i.e. the depot "replicated"); and in between are the customers.

$VI := \{1, \cdots, n + 1\}$: the initial depot and the $n$ customers.

$VF := \{2, \cdots, n + 2\}$: the customers and the final depot.

$VC := \{2, \cdots, n + 1\}$: customers.

$T$: the days considered in the planning horizon.

$K$: available vehicles.

$IT$: time intervals that make up a day.

$IT^-$: $IT$ excluding the last element.

$VC_f$: customers to be visited with frequency $f$.

$P_f$: possible visiting patterns to customers in $VC_f$.

### Vectorial parameters

$d_{ij}$: distance between nodes $i$ and $j$ when considering time dependent models. It denotes the travel time between nodes $i$ and $j$ when considering models without time dependency.

$b_i$: number of visits in the period to the customer $i$.

$a_p^l$: 1 if day $l$ is included in visit pattern $p$, 0 otherwise.

$\tau_i$: service time at node $i$.

$e_i$: left end of the time window on customer $i$.

$r_i$: right end of the time window on customer $i$.

$\theta_u$: the left-end of time interval $u$.

$v_u$: the considered standard velocity in the time interval $u$.

**Decision variables**

$x_{ij}^l$: 1 if node $j$ is visited after node $i$ on day $l$, 0 otherwise.

$w_{ik}$: 1 if customer or initial depot $i$ is visited by vehicle $k$ over the entire planning horizon.

$t_{ik}^l$: The time at which service starts at customer $i$ on the day $l$ by vehicle $k$.

$y_{ip}$: 1 if customer $i$ is assigned to visit pattern $p$, 0 otherwise.

$s_{iu}^l$: 1 if the service of customer $i$ is finished on day $l$ in the interval $u$, 0 otherwise.

$zmaxi$: the maximum duration of a route.

$zcustmaxi$: the maximum time at which a customer is visited.

**Objective functions**

$$f1: \quad \min z = zmaxi. \tag{1}$$

$$f2: \quad \min z = zcustmaxi. \tag{2}$$

$$f3: \quad \min z = \sum_{i \in VI} \sum_{j \in VF} \sum_{l \in T} d_{ij} \cdot x_{ij}^l. \tag{3}$$

**Constraints**

$$\sum_{p \in P_f} y_{jp} = 1, \quad \forall j \in VC_f, \quad f = 1, \cdots, p\_num. \tag{4}$$

$$\sum_{j \in VF, j \neq i} x_{ij}^l = \sum_{p \in P_f} (a_p^l \cdot y_{ip}), \forall i \in VC_f, \quad f = 1, \cdots, p\_num \quad \forall l \in T. \tag{5}$$

$$\sum_{j \in VC} x_{1j}^l \leq m, \quad \forall l \in T. \tag{6}$$

$$\sum_{j \in VC} x_{1j}^l = \sum_{i \in VC} x_{i,n+2}^l, \forall \, l \in T. \tag{7}$$

$$\sum_{j \in VI, i \neq j} x_{ji}^l = \sum_{j \in VF, i \neq j} x_{ij}^l, \quad \forall i \in VC, \quad \forall l \in T. \tag{8}$$

$$\sum_{k \in K} w_{ik} = 1, \quad \forall i \in VC. \tag{9}$$

$$w_{1k} = 1, \quad \forall k \in K. \tag{10}$$

$$\sum_{l \in T} x_{ij}^l \leq b_i \cdot (1 - w_{ik} + w_{jk}), \quad \forall i,j \in VC, \quad \forall k \in K. \tag{11}$$

$$x_{1i}^l + x_{1j}^l \leq 3 - (w_{ik} + w_{jk}), \quad \forall i,j \in VC, \quad i \neq j, \quad \forall l \in T, \quad \forall k \in K. \tag{12}$$

$$x_{ii}^l = 0, \ \forall i \in VC, \quad \forall l \in T. \tag{13}$$

$$t_{1k}^l = 0, \ \forall \ l \in T, \quad \forall k \in K. \tag{14}$$

$$t_{ik}^l \geq \boldsymbol{e}_i, \ \forall i \in VC, \quad \forall l \in T, \quad \forall k \in K. \tag{15}$$

$$t_{ik}^l \leq \boldsymbol{r}_i - \boldsymbol{\tau}_i, \ \forall i \in VC, \quad \forall l \in T, \quad \forall k \in K. \tag{16}$$

$$t_{ik}^l \geq \boldsymbol{d}_{1i} - \boldsymbol{D} \cdot (2 - x_{1i}^l - w_{ik}), \ \forall i \in VC, \quad \forall l \in T, \quad \forall k \in K. \tag{17}$$

$$t_{ik}^l \geq (\boldsymbol{d}_{1i}/\boldsymbol{v}_1) - \boldsymbol{D} \cdot (2 - x_{1i}^l - w_{ik}), \ \forall i \in VC, \quad \forall l \in T, \quad \forall k \in K. \tag{18}$$

$$t_{ik}^l + \boldsymbol{\tau}_i + \boldsymbol{d}_{ij} \leq t_{jk}^l + \boldsymbol{D} \cdot (1 - x_{ij}^l), \forall i, j \in VC, \quad \forall l \in T, \quad \forall k \in K. \tag{19}$$

$$t_{ik}^l + \boldsymbol{\tau}_i + \boldsymbol{d}_{ij} \leq t_{jk}^l + \boldsymbol{D} \cdot (1 - x_{ij}^l), \forall i \in VC, \quad \forall j \in VF, \quad \forall l \in T, \quad \forall k \in K. \tag{20}$$

$$\sum_{u \in IT} s_{iu}^l \leq 1, \ \forall i \in VC, \quad \forall l \in T. \tag{21}$$

$$\sum_{u \in IT} s_{iu}^l \geq \sum_{j \in VI} x_{ji}^l, \ \forall i \in VC, \quad \forall l \in T. \tag{22}$$

$$t_{ik}^l + \boldsymbol{\tau}_i \leq \boldsymbol{\theta}_{u+1} - \epsilon + \boldsymbol{D} \cdot (1 - s_{iu}^l) + \boldsymbol{D} \cdot (1 - \sum_{j \in VF} x_{ij}^l),$$
$$\forall i \in VC, \quad \forall u \in IT^-, \quad \forall l \in T, \quad \forall k \in K. \tag{23}$$

$$t_{ik}^l + \boldsymbol{\tau}_i \geq \boldsymbol{\theta}_u - \boldsymbol{D} \cdot (1 - s_{iu}^l) - \boldsymbol{D} \cdot (1 - \sum_{j \in VF} x_{ij}^l),$$
$$\forall i \in VC, \quad \forall u \in IT, \quad \forall l \in T, \quad \forall k \in K. \tag{24}$$

$$t_{ik}^l + \boldsymbol{\tau}_i + \sum_{u \in IT} (\boldsymbol{d}_{ij}/\boldsymbol{v}_u) \cdot s_{iu}^l \leq t_{jk}^l + \boldsymbol{D} \cdot (1 - x_{ij}^l),$$
$$\forall \ i, j \in VC, \quad \forall l \in T, \quad \forall k \in K. \tag{25}$$

$$t_{n+2,k}^l \leq zmaxi, \forall l \in T, \quad \forall k \in K. \tag{26}$$

$$t_{il}^l \leq zcustmaxi, \forall i \in VC, \quad \forall l \in T, \quad \forall k \in K. \tag{27}$$

$$x_{1i}^l + x_{i,n+2}^l \leq 1, \ \forall i \in VC, \quad \forall l \in T. \tag{28}$$

$$w_{n+1,1} = 1. \tag{29}$$

$$x_{ij}^l + x_{ji}^l \leq 1, \ \forall i, j \in V, \quad i \neq j \quad \forall l \in T. \tag{30}$$

$$x_{ij}^l \le 1 - w_{ik} + w_{jk}, \quad \forall i, j \in VC, \quad \forall l \in T, \quad \forall k \in K. \tag{31}$$

$$x, w, y, s \in \{0, 1\}, \quad t, zmaxi, zcustmaxi \ge 0. \tag{32}$$

The following are the valid constraints for all models:

1. One pattern per customer Eq (4).

2. Any customer should be visited in all days of his pattern assigned Eq (5).

3. The number of routes on any day should be at most the number of vehicles available Eq (6).

4. The number of routes departing from the initial depot is equal to the number of routes arriving to the final depot Eq (7).

5. Balance at a node: If a route arrives to a node, it must leave from this node to another one (perhaps the depot) Eq (8).

6. One vehicle should be assigned to each customer Eq (9).

7. The initial depot is assigned to all available vehicles Eq (10).

8. A vehicle can go from one customer to another if both customers are assigned to it Eq (11).

9. A vehicle cannot go any day from the initial depot to more than one customer associated with this vehicle Eq (12).

10. A route cannot go from a customer to the same one. Eq (13).

11. Constraint Eq (32) specifies the type of variables.

The following are additional constraints that appear in the variants with time windows: The beginning time of any route is zero Eq (14), the service start time on any customer must be at least the lower end of its respective time window Eq (15), the service end time on any customer must be less or equal than the upper end of its respective time window Eq (16).

The arrival time at the first customer of any route is at least the time it takes to travel the first edge Eq (17) or Eq (18).

1. In the case without TD, $d_{1i}$ represents the time to go from the depot to customer $i$ Eq (17).

2. In the case with TD, $d_{1i}$ represents the distance between the depot and customer $i$ Eq (18).

For the subtours elimination constraints (some from constraints (19) to (25)) we distinguish between the following cases:

1. Case without TD:

(a) With objective function $f2$: If the vehicles goes from node $i$ to $j$, it is not possible to start service in $j$ before the time of reaching node $i$ plus service time in $i$ plus travel time from $i$ to $j$ Eq (19).

(b) With objective function $f1$: The adequate constraint instead of (19) is (20), where the domain of $j$ is $VF$, including the node $n + 2$ (final depot).

2. Case with TD. The constraints are: (21–25)

(a) A node $i$ is reached on a day $l$ during a single time interval $u$ Eqs (21) and (22).

(b) The vehicles' speed on the next route section is determined according with the time interval at which the service was concluded in the previously visited customer $i$ Eqs (23) and (24).

(c) If the vehicles goes from node $i$ to $j$, it is not possible to start service in $j$ before the time for reaching node $i$ plus service time in $i$ plus travel time from $i$ to $j$ Eq (25). It is the variant of the constraint (19) when considering TD.

The specific constraints associated with objective functions are: With $f1$: (26), and with $f2$: (27). When using objective function $f3$, it is not necessary to include the constraints (26) and (27) in the model.

The following constraints can be added to any model: constraint (28) prevents a route from having a single customer as in [12]. The constraint (29) is a possible strategy for breaking the symmetry inherent in the PVRP definition, previously used in [12]. The constraint (30) is a valid inequality which indicates that on any given day an edge can only be traversed in one direction.

Note that the sum of constraints in (31) originates (11), so the set of constraints given by (31) can be used in any model instead constraint (11). If this is the case, we are talking about a disaggregated version of the model [53].

## Experimental setup

This section describes the characteristics of the numerical experimentation developed to analyze the behavior of the ConPVRP-TW, with the three objective functions, as described in the modeling framework. This study will serve as a basis for future research questions concerning the performance of other models considered in the framework that involve greater computational complexity. To analyze the ConPVRP-TW three factors are varied: frequency of visits per customer, centrality of the depot, and the objective function (i.e. $f1$, $f2$ and $f3$). First, we describe the instances generated to represent practical PVRP problems and then the design of the experiments that was later conducted.

### Instances description

We chose to build our own data set, given that there are no data sets reported in the literature that allows explicitly the evaluation of the effect of the depot centrality, and as observed in [12], the classical data for PVRP are highly symmetric in terms of the spatial distribution of the nodes and the allowed visit schedules of the customers, which generates solutions to the PVRP that are already driver consistent.

The horizon of the instances analyzed is a week. We consider two different frequencies: weekly and semiweekly. The instance size, which is denoted it with $\omega$, is determined by the total number of visits during the planning horizon. Thus, sets of $\omega$ positions were quasi-randomly generated to locate the customers. For a given $\omega$, the positions were used in instances, with three different visit frequencies each, and therefore, different number of customers, in this way: Instances with weekly frequency (W) use sets of $n = \omega$ customer positions; instances with semiweekly frequency (S) use a subset of $n = \omega/2$ customer positions, and instances with a mix of both frequencies (M) use about a half of the positions for each frequency. To clarify the last case, let's take as an example the mixed frequency with $\omega = 30$ visits: $\omega/2 - 1 = 14$ visits correspond to 14 customers with weekly frequency, and the remaining 16 visits correspond to 8 customers with semiweekly frequency, for a total of $n = 22$ customers. Similarly, for $\omega = 20$ visits, $\omega/2 = 10$ visits correspond to 10 customers with weekly frequency and the remaining 10 visits correspond to 5 customers with biweekly frequency, for a total of $n = 15$ customers.

Finally, the sets were duplicated and a central depot (C) and an outer depot (NC) were added to each one.

Three groups were generated with a total of eight data sets in square areas chosen arbitrarily. The customer locations were randomly assigned following operational circumstances consistent to the real problem addressed. For example, Group A instances seek to include situations where distribution is over suburban areas, comprising a relatively larger area and a low density of customers. In contrast, Group B instances are intended to reflect cases of "last mile distribution" in which zoning involves small distribution areas and varying customer density over a wider range. Two data sets conform Group C that uses Group A setting to test the effect of the number of visits $\omega$ when the number of customers $n$ is fixed. Group A has 2 data sets with the origin (0, 0) as a vertex of the rectangle: the first one with $n$ = 20 customers located in the box (0, 200) × (0, 100) and a second set with $n$ = 30 customers located in the box (0, 100) × (0, 120). Group B has four sets with $n$ = 12, 20, 30, and 40 customers each, randomly located in the box (−4, 4) × (−4, 4), with the origin (0, 0) in the center of the rectangle. Tables 5 and 6 show the instance structure for these six data sets. The first set in Group C is the same set with $n$ = 20 customers from Group A, see Fig 1(d), and the second one has $n$ = 16 customers located in the same area of the Group A's second set as shown in Fig 2.

Fig 1(a), 1(b) and 1(c) show the locations of $n$ = 30, $n$ = 22 and $n$ = 15 customers respectively, all of them used in instances with $\omega$ = 30 visits, with weekly, mixed and semiweekly frequencies, respectively, being the configurations shown in Fig 1(b) and 1(c) subsets of the configuration shown in Fig 1(a). Fig 1(d), 1(e) and 1(f) show the locations with $n$ = 20, $n$ = 15 and $n$ = 10 customers; all of them used in instances with $\omega$ = 20 visits, with weekly, mixed and semiweekly frequencies, respectively, being the last two configurations subsets of the configuration shown in Fig 1(d). Fig 1 also identifies the position of the depot for the runs with centered depot (C), with a red bullet. For runs with outer depot, the location used for the depot was the origin.

Fig 3(a) to 3(d) show the set of four generated data sets that belong to Group B, with $n$ = 60, $n$ = 40, $n$ = 20 and $n$ = 12 customers, respectively. These locations are used in instances with weekly frequency (W), and from each of them two subsets are randomly extracted to form instances with semiweekly (S) and mixed (M) frequencies with the number of customers specified in Table 6. For instances with centered depot, the depot coordinates are given by the geometric median of the customer's locations, while for instances with outer depot, their coordinates were selected so that the centrality is 10%.

Table 5. Instance structure Group A.

| Number of visits ($\omega$) | 30 | | | 20 | | |
|---|---|---|---|---|---|---|
| Frequency | W | S | M | W | S | M |
| Number of customers ($n$) | 30 | 15 | 22 | 20 | 10 | 15 |

W: weekly, S: Semiweekly, M: Mixed

Table 6. Instance structure Group B.

| Number of visits ($\omega$) | 60 | | | 40 | | | 20 | | | 12 | | |
|---|---|---|---|---|---|---|---|---|---|---|---|---|
| Frequency | W | S | M | W | S | M | W | S | M | W | S | M |
| Number of customers ($n$) | 60 | 30 | 45 | 40 | 20 | 30 | 20 | 10 | 15 | 12 | 6 | 9 |

W: weekly, S: Semiweekly, M: Mixed

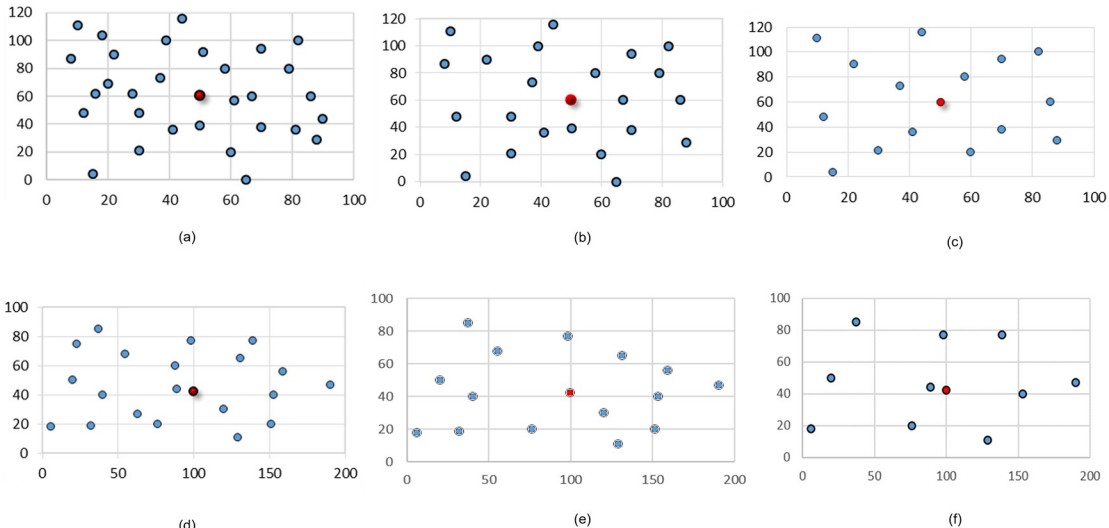

**Fig 1. Spatial distribution of customers—Group A—Centered depot.**

Other parameters used in the model are: $M = 2$ vehicles, $T = 5$ days, $\tau(i) = 10 \ \forall i$, $r_i = 120 \ \forall i$. $D = 150$ for Groups A and C instances with centered depot and $D = 180$ for instances with outer depot. For all Group B instances $D = 50$.

All the information required for the replication of the experiments is housed in the paper repository https://github.com/jamartinec/Data_Paper_Baldoquin_Diaz_Martinez.

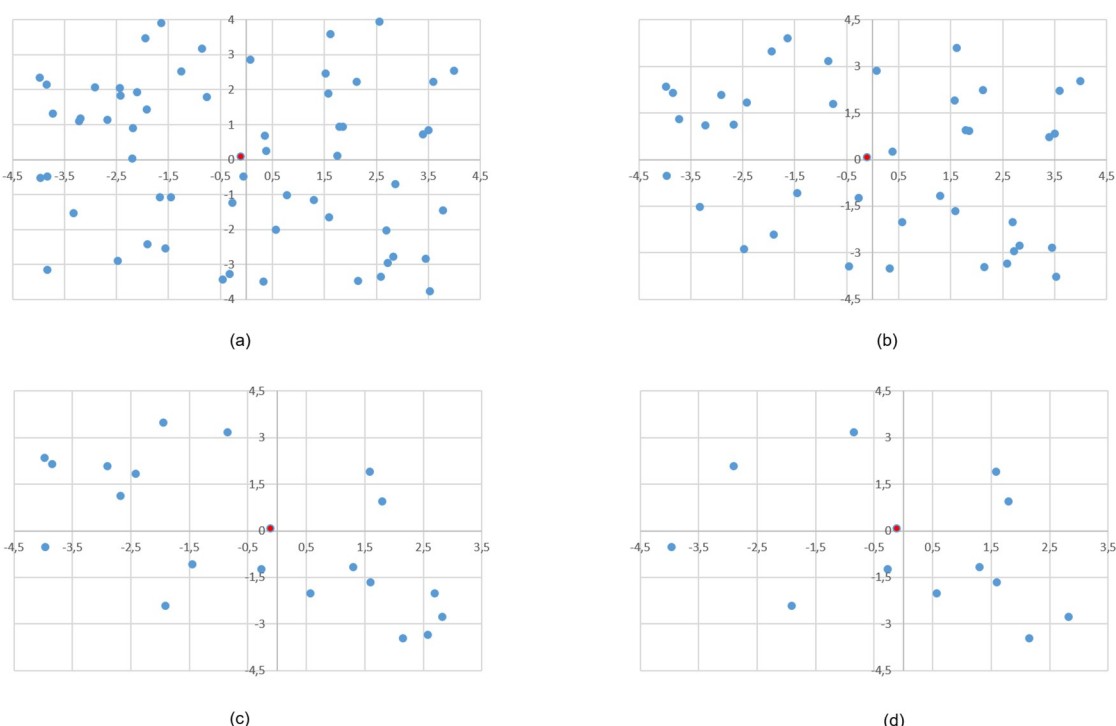

**Fig 2. Spatial distribution of $n = 16$ customers—Group C—Centered depot.**

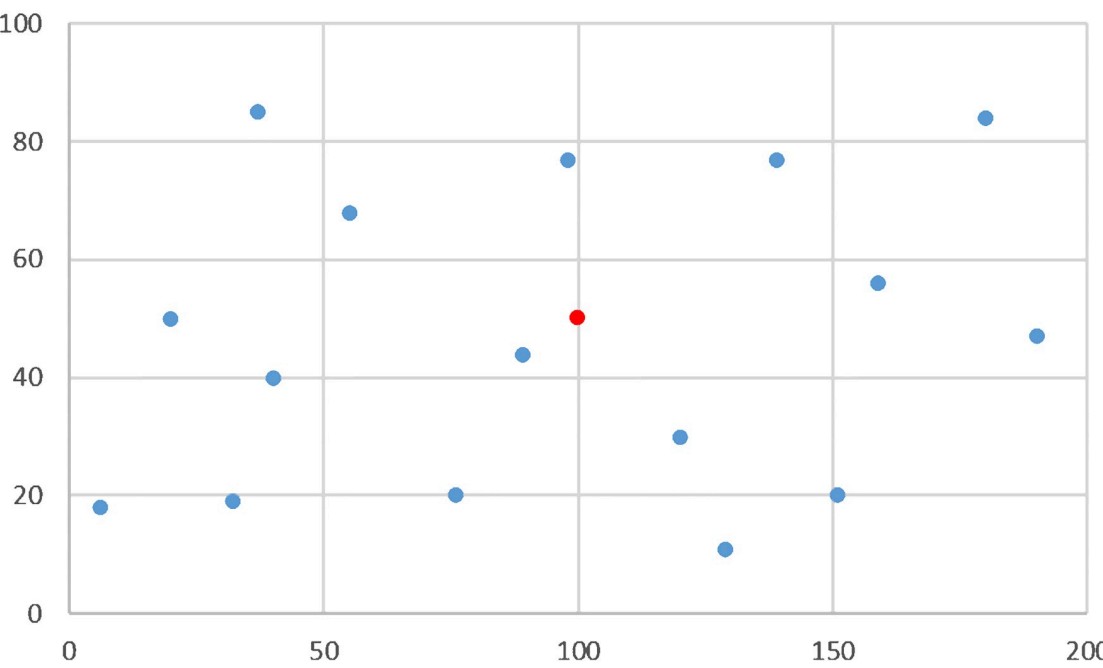

**Fig 3. Spatial distribution of customers—Group B—Centered depot.**

## Experimental design

With the aim of assessing the performance of the model framework discussed above, we conducted the experimental design described in Table 7.

The factors considered are: the frequency, the optimization function, and depot position. This means that a complete experimentation for a given size involves 18 runs. We present in this paper the complete experimentation for the six data sets shown in the Tables 5 and 6.

We also study the effect of using the constraint (28) which prevents a route of having a single customer, the constraint for symmetry breakage (29), the valid inequalities given by (30) which states that and edge can only be traversed in one direction on the same route, and the influence of using the set of disaggregated constraints (31) instead constraint (11).

## Performance measures

In this work, we solved the different instances that we have just described using the general-purpose MILP solver Gurobi-8.1.1. The modeling language used was Pyomo. The experiments were run in a computer with 8 CPUs Intel® Xeon® E5-2670 2.60GHz, operating system Linux Rocks 6.2. Under the same parameter tuning of the solver, modeling performance is assessed with two values: the objective function value (OF) and the relative gap (%). Table 8 shows the values used for some of the most important parameters that can influence the solution process of a MILP model [54].

**Table 7. Experimental design for each number of visits.**

| Factor | Options | | |
|---|---|---|---|
| Frequency | Weekly | Semiweekly | Mixed |
| Optimization function | f1 | f2 | f3 |
| Depot position | Center (C) | Outer (NC) | |

**Table 8. Gurobi parameter tunning.**

| Parameter | Description | Value | Effect |
|---|---|---|---|
| Threads | Controls the number of threads used by the parallel MIP solver | Default: 0 | Use all the cores available in the machine |
| MIPFocus | Allows to decide which aspects to prioritize between finding new feasible solutions and proving that current solution is optimal | 3 | Focuses on improving the best bound |
| TimeLimit | Limits the runtime | 3600 | Stops the solution process after 1 hour of starting the algorithm |
| MIPGap | Relative MIP optimality gap | Default: 1e-4 | The solver will therminate when the absolute gap is less than MIPGap |

The B&B algorithms used by the solver to solve MIP problems keep the best integer solution found along with its objective function value $\hat{z}$, this is called the incumbent solution. If the problem is a minimization one, $\hat{z}$ is an upper bound for the optimal solution of the original MILP formulation. Further details of the B&B algorithms can be found in [55]. At any time during the B&B search, there is also a lower valid bound, called the best bound $z^*$, which is obtained by taking the minimum of the optimal values of the objective function on the leaf nodes. The absolute gap is the difference between the bounds, i.e. $gap = \hat{z} - z^*$. The relative gap is obtained by dividing the absolute gap by the best lower bound, that is $rel\_gap = gap/z^*$. When the gap is less than a small value $\epsilon$, the incumbent solution is returned as the optimal for the original problem [54, 56].

The choice of the relative gap as the main performance measure is consistent with Klotz & Newman [57], who showed that through careful formulation and algorithmic parameter tuning, the optimizer performance can be improved in terms of the optimality gaps (%). In addition, recognizing that commercial solvers are largely a black box, studying the effect of certain model constraints, remains a valid research question [58]. On other hand, the OF serves to compare variants of the same model, and is the typical performance measure of optimization models (for example, [59, 60]). Results reported are the values obtained after a fixed maximum computation time of one hour. Time selection obeyed to a preliminary experimentation with 10-hour runs, where it could be observed that the best lower bounds where achieved in this period, and following [57], good lower bounds better reflect the difficulty of a model solution than other aspects of the solution process.

## Results

This section shows the results of the experimental runs for Groups A, B and C, in Tables 9 to 16, followed by a brief reading of the most outstanding values. Statistical analysis and further discussion is presented in the following section.

Tables 9 and 10 report the experimental results obtained by using Group A instances, and Tables 11, 12, 13 and 14 report the results by using Group B instances. The experiments results in which the frequency is varied given a fixed number of customers $n$, are recorded in Tables 15 for $n = 20$ and 16 for $n = 16$. Each line in the tables specifies the objective function, or the combination of objective function and additional constraint that has been tested.

Table 9 shows that optimality is reached only with instances with objective function *f*3 and variant *f*3 + (31), centered depot and $n = 15$ customers. In general, *f*3 and variant *f*3 + (31) present better gap values than instances with *f*2, and *f*2 + (31). *f*1 instances have 100% gap which usually means that the solver has not yet computed a lower bound obtained as optimal solution for a linear programming relaxation. No consistent effect of the additional constraints is observed.

**Table 9. Experimental results $\omega$ = 30 visits (Group A).**

| Visits ($\omega$) | 30 | | | | | | | | | | | |
|---|---|---|---|---|---|---|---|---|---|---|---|---|
| Depot | C | | | | | | NC | | | | | |
| Frequency | W ($n$ = 30) | | M ($n$ = 22) | | S ($n$ = 15) | | W ($n$ = 30) | | M ($n$ = 22) | | S ($n$ = 15) | |
| | OF | % | OF | % | OF | % | OF | % | OF | % | OF | % |
| f1 | 41,00 | 100,00 | 48,00 | 100,00 | 54,00 | 100,00 | 54,00 | 100,00 | 56,00 | 100,00 | 63,00 | 100,00 |
| f2 | 38,00 | 78,90 | 38,00 | 66,30 | 46,00 | 82,60 | 37,00 | 64,90 | 41,99 | 61,90 | 46,00 | 71,70 |
| f3 | 80,00 | 17,50 | 110,00 | 10,00 | 108,00 | 0,00 | 121,00 | 46,28 | 136,00 | 27,94 | 136,00 | 19,85 |
| f1 + (28) | 48,00 | 100,00 | 43,00 | 100,00 | 49,00 | 100,00 | 52,00 | 100,00 | 47,00 | 100,00 | 63,00 | 100,00 |
| f2 + (28) | 37,00 | 90,00 | 39,00 | 64,10 | 46,00 | 65,22 | 39,00 | 61,54 | 47,00 | 70,22 | 48,00 | 68,53 |
| f1 + (30) | 45,00 | 100,00 | 44,00 | 100,00 | 52,00 | 100,00 | 51,00 | 100,00 | 68,00 | 100,00 | 62,00 | 100,00 |
| f2 + (30) | 36,00 | 77,77 | 41,00 | 68,29 | 44,00 | 72,72 | 38,99 | 66,66 | 33,00 | 59,46 | 47,00 | 70,21 |
| f1 + (31) | 39,00 | 100,00 | 44,00 | 100,00 | 48,00 | 100,00 | 50,00 | 100,00 | 57,00 | 100,00 | 61,00 | 100,00 |
| f2 + (31) | 36,00 | 77,77 | 38,59 | 67,81 | 42,99 | 69,76 | 43,00 | 69,76 | 45,00 | 64,44 | 46,99 | 70,21 |
| f3 + (31) | 78,00 | 15,38 | 113,00 | 2,65 | 108,00 | 0,00 | 116,00 | 43,95 | 149,00 | 29,53 | 136,00 | 22,79 |

OF: objective function value; %: relative gap.

Table 10 shows that instances with centered depot and objective functions $f2$ and $f3$ get to the optimal or near, except for mixed frequency cases. Objective function values for $f1$ have still 100% gap, except for some cases with $n$ = 10. $f3$ instances with non-centered depot behave better with constraint (31) than without it. In general, gaps with $f3$ and $f3$ + (31) behave better than with $f2$ and $f2$ + (31).

Tables 11 and 12 show that with 60 and 40 visits, objective function $f3$ and variant $f3$ + (31) have a better performance than the other variants, although none of them reaches the optimum. The value of the gap for $f3$ and its variants behave better for centered instances compared to the non-centered ones. Only the additional constraint (30) for 60 visits instances, and the constraint (31) for 40 visits instances have slightly but consistent positive effect in $f2$ gaps.

**Table 10. Experimental results $\omega$ = 20 visits (Group A).**

| Visits ($\omega$) | 20 | | | | | | | | | | | |
|---|---|---|---|---|---|---|---|---|---|---|---|---|
| Depot | C | | | | | | NC | | | | | |
| Frequency | W ($n$ = 20) | | M ($n$ = 15) | | S ($n$ = 10) | | W ($n$ = 20) | | M ($n$ = 15) | | S ($n$ = 10) | |
| | OF | % | OF | % | OF | % | OF | % | OF | % | OF | % |
| f1 | 33,00 | 100,00 | 45,00 | 100,00 | 41,00 | 39,00 | 49,00 | 100,00 | 57,00 | 100,00 | 56,00 | 100,00 |
| f2 | 21,00 | 0,00 | 32,00 | 53,10 | 34,00 | 3,00 | 30,00 | 23,00 | 38,00 | 40,00 | 35,99 | 3,00 |
| f3 | 67,00 | 0,00 | 113,00 | 0,00 | 120,00 | 0,00 | 89,00 | 22,47 | 148,00 | 27,02 | 150,00 | 13,33 |
| f1 + (28) | 34,00 | 100,00 | 45,00 | 100,00 | 41,00 | 38,00 | 56,00 | 100,00 | 57,00 | 100,00 | 56,00 | 100,00 |
| f2 + (28) | 21,00 | 0,00 | 32,00 | 43,00 | 33,00 | 0,00 | 33,00 | 30,00 | 41,00 | 44,00 | 36,00 | 0,00 |
| f1 + (30) | 33,00 | 100,00 | 41,00 | 100,00 | 41,00 | 80,00 | 49,00 | 100,00 | 58,00 | 100,00 | 56,00 | 46,00 |
| f2 + (30) | 21,00 | 0,00 | 32,00 | 66,00 | 34,00 | 0,00 | 30,00 | 23,00 | 38,00 | 39,00 | 36,00 | 3,70 |
| f1 + (31) | 34,00 | 100,00 | 44,00 | 100,00 | 41,00 | 100,00 | 49,00 | 100,00 | 58,00 | 100,00 | 56,00 | 100,00 |
| f2 + (31) | 21,00 | 0,00 | 33,00 | 61,00 | 34,00 | 0,00 | 30,00 | 47,00 | 40,00 | 53,00 | 36,00 | 0,00 |
| f3 + (31) | 67,00 | 0,00 | 113,00 | 0,00 | 120,00 | 0,00 | 90,00 | 23,33 | 78,00 | 0,00 | 150,00 | 0,00 |

OF: objective function value; %: relative gap.

**Table 11. Experimental results $\omega$ = 60 visits (Group B).**

| Visits ($\omega$) | 60 | | | | | | | | | | | |
|---|---|---|---|---|---|---|---|---|---|---|---|---|
| Depot | C | | | | | | NC | | | | | |
| Frequency | W ($n$ = 60) | | M ($n$ = 45) | | S ($n$ = 30) | | W ($n$ = 60) | | M ($n$ = 45) | | S ($n$ = 30) | |
| | OF | % | OF | % | OF | % | OF | % | OF | % | OF | % |
| f1 | 37,34 | 100,00 | 29,15 | 100,00 | 32,55 | 100,00 | 37,60 | 100,00 | 30,41 | 100,00 | 32,68 | 100,00 |
| f2 | 21,16 | 94,25 | 28,32 | 94,36 | 22,34 | 85,17 | 21,17 | 92,42 | 33,02 | 95,25 | 27,51 | 87,82 |
| f3 | 68,01 | 28,04 | 95,11 | 30,78 | 82,22 | 10,99 | 90,36 | 45,57 | 118,15 | 45,57 | 97,27 | 25,68 |
| f1 + (28) | 24,53 | 100,00 | 37,00 | 100,00 | 25,89 | 100,00 | 40,33 | 100,00 | 30,07 | 100,00 | 33,55 | 100,00 |
| f2 + (28) | 21,45 | 91,76 | 29,56 | 95,25 | 24,47 | 85,26 | 22,90 | 91,22 | 31,06 | 94,67 | 26,72 | 83,44 |
| f1 + (30) | 28,85 | 100,00 | 26,11 | 100,00 | 36,75 | 100,00 | 37,40 | 83,10 | 30,45 | 100,00 | 34,83 | 100,00 |
| f2 + (30) | 20,35 | 90,12 | 23,27 | 88,01 | 24,30 | 81,93 | 20,19 | 88,50 | 31,37 | 94,76 | 24,65 | 82,14 |
| f1 + (31) | 26,03 | 100,00 | 28,33 | 100,00 | 32,29 | 100,00 | 29,50 | 100,00 | 29,76 | 100,00 | NAN | NAN |
| f2 + (31) | 25,88 | 94,43 | 29,15 | 100,00 | 20,91 | 74,02 | 22,21 | 91,11 | 30,15 | 95,98 | 29,19 | 79,50 |
| f3 + (31) | 69,50 | 29,27 | 82,52 | 21,67 | 80,68 | 8,53 | 98,27 | 50,06 | 124,03 | 48,79 | 94,80 | 23,42 |

OF: objective function value; %: relative gap.

Table 13 shows that optimality of $f1$ and $f1$ + (31) instances is reached for some instances with $n$ = 10 customers and semiweekly frequency. It is also observed that $f2$ and $f3$ gaps behave better for centered instances compared to the non-centered ones. Centered instances with $f2$, $f3$, and their variants get to the optimal except for most of the $f2$ cases for which the number of customers was higher ($n$ = 20). No consistent improvements for the addition of constraints is observed.

In Table 14 it can be seen that $f1$ and instances weekly frequencies still have 100% gaps. Additional constraints do not improve any objective function. The remaining instances get to optimality, except for one $f2$ instance with the larger value of $n$ = 12, weekly frequency and non-centered depot.

In Tables 15 and 16, for the established number of customers, $f1$ instances still have 100% gaps. With a fixed $n$, the higher the number of visits ($\omega$) the larger the gaps for $f2$ and $f3$

**Table 12. Experimental results $\omega$ = 40 visits (Group B).**

| Visits ($\omega$) | 40 | | | | | | | | | | | |
|---|---|---|---|---|---|---|---|---|---|---|---|---|
| Depot | C | | | | | | NC | | | | | |
| Frequency | W ($n$ = 40) | | M ($n$ = 30 | | S ($n$ = 20) | | W ($n$ = 40) | | M ($n$ = 30) | | S ($n$ = 20) | |
| | OF | % | OF | % | OF | % | OF | % | OF | % | OF | % |
| f1 | 17,97 | 100,00 | 18,01 | 100,00 | 20,46 | 100,00 | 22,58 | 100,00 | 24,23 | 100,00 | 24,39 | 100,00 |
| f2 | 13,58 | 80,12 | 14,55 | 78,89 | 12,78 | 71,13 | 15,58 | 84,20 | 15,99 | 79,19 | 15,41 | 77,22 |
| f3 | 49,37 | 14,19 | 66,62 | 18,75 | 70,92 | 13,32 | 66,50 | 37,56 | 81,11 | 34,22 | 81,26 | 25,26 |
| f1 + (28) | 18,06 | 100,00 | 17,96 | 100,00 | 19,13 | 100,00 | 28,13 | 100,00 | 25,66 | 100,00 | 24,29 | 100,00 |
| f2 + (28) | 14,29 | 81,11 | 15,50 | 76,90 | 15,03 | 72,85 | 16,25 | 83,38 | 19,14 | 83,44 | 16,61 | 72,13 |
| f1 + (30) | 17,02 | 100,00 | 17,22 | 100,00 | 21,70 | 100,00 | 29,51 | 100,00 | 24,32 | 100,00 | 25,14 | 100,00 |
| f2 + (30) | 17,95 | 86,30 | 17,69 | 81,54 | 18,47 | 74,61 | 16,44 | 82,00 | 18,82 | 83,53 | 16,10 | 72,36 |
| f1 + (31) | 16,46 | 100,00 | 17,34 | 100,00 | 20,97 | 100,00 | 23,11 | 100,00 | 24,03 | 100,00 | 26,44 | 100,00 |
| f2 + (31) | 11,81 | 77,13 | 16,71 | 72,47 | 15,13 | 70,06 | 15,24 | 82,30 | 17,55 | 74,81 | 18,67 | 71,14 |
| f3 + (31) | 49,37 | 15,45 | 65,42 | 18,14 | 70,54 | 14,47 | 57,97 | 28,12 | 86,18 | 37,87 | 76,98 | 20,97 |

OF: objective function value; %: relative gap.

**Table 13. Experimental results ω = 20 visits (Group B).**

| Visits (ω) | 20 | | | | | | | | | | | |
|---|---|---|---|---|---|---|---|---|---|---|---|---|
| Depot | C | | | | | | NC | | | | | |
| Frequency | W (n = 20) | | M (n = 15) | | S (n = 10) | | W (n = 20) | | M (n = 15) | | S (n = 10) | |
| | OF | % | OF | % | OF | % | OF | % | OF | % | OF | % |
| f1 | 11,05 | 100,00 | 11,04 | 100,00 | 11,58 | 100,00 | 19,38 | 100,00 | 20,38 | 100,00 | 20,25 | 0,00 |
| f2 | 5,63 | 5,33 | 5,93 | 0,00 | 6,57 | 0,00 | 9,62 | 65,49 | 10,03 | 6,08 | 10,26 | 0,00 |
| f3 | 29,86 | 0,00 | 38,72 | 0,00 | 46,56 | 0,00 | 37,67 | 20,28 | 47,96 | 14,97 | 50,52 | 0,00 |
| f1 + (28) | 11,79 | 100,00 | 12,17 | 100,00 | 11,58 | 100,00 | 20,38 | 100,00 | 20,38 | 100,00 | 20,45 | 100,00 |
| f2 + (28) | 6,70 | 0,00 | 6,93 | 0,00 | 7,57 | 0,00 | 11,19 | 45,67 | 11,19 | 0,00 | 11,26 | 0,00 |
| f1 + (30) | 11,00 | 100,00 | 12,17 | 100,00 | 11,58 | 100,00 | 19,38 | 100,00 | 20,05 | 100,00 | 20,28 | 100,00 |
| f2 + (30) | 6,63 | 4,52 | 6,93 | 0,00 | 7,57 | 0,00 | 10,19 | 57,61 | 11,03 | 2,54 | 11,26 | 0,00 |
| f1 + (31) | 10,73 | 100,00 | 11,04 | 100,00 | 11,58 | 0,00 | 19,38 | 100,00 | 20,45 | 100,00 | 20,25 | 0,00 |
| f2 + (31) | 6,63 | 2,56 | 6,93 | 0,00 | 7,57 | 0,00 | 10,19 | 57,61 | 10,75 | 2,54 | 11,26 | 0,00 |
| f3 + (31) | 29,86 | 0,00 | 38,72 | 0,00 | 46,56 | 0,00 | 37,67 | 22,23 | 47,96 | 15,59 | 50,52 | 0,00 |

OF: objective function value; %: relative gap.

instances. For *f*3 instances, constraint (31) has no effect in the gaps obtained. However, constraints (28) and (30) have consistently slight improvements in the *f*2 gaps.

## Analysis and discussion

Having into account that the maximum computation time allowed was 1 hour, next we present a series of highlights that come out from the results obtained.

Concerning to the additional constraints (28), (30), and (31), statistical two-sample difference tests were performed to verify if the gap improved when adding the additional constraints, one at a time. Every test considered *n* = 6, with all combinations of centered and non-centered and frequency for each number of visits. *p*_values of the tests are shown in Table 17. Detailed information of the statistical tests can be found in S1 Appendix. With an *α* = 0.05, the

**Table 14. Experimental results ω = 12 visits (Group B).**

| Visits (ω) | 12 | | | | | | | | | | | |
|---|---|---|---|---|---|---|---|---|---|---|---|---|
| Depot | C | | | | | | NC | | | | | |
| Frequency | W (n = 12) | | M (n = 9) | | S (n = 6) | | W (n = 12) | | M (n = 9) | | S (n = 6) | |
| | OF | % | OF | % | OF | % | OF | % | OF | % | OF | % |
| f1 | 9,38 | 100,00 | 9,18 | 0,00 | 9,18 | 0,00 | 19,02 | 100,00 | 19,00 | 0,00 | 15,90 | 0,00 |
| f2 | 4,19 | 0,00 | 4,34 | 0,00 | 4,34 | 0,00 | 9,01 | 0,00 | 9,01 | 0,00 | 7,45 | 0,00 |
| f3 | 24,15 | 0,00 | 35,05 | 0,00 | 32,02 | 0,00 | 26,47 | 0,00 | 44,08 | 0,00 | 38,16 | 0,00 |
| f1 + (28) | 10,98 | 100,00 | 11,79 | 0,00 | 10,18 | 0,00 | 20,20 | 100,00 | 20,31 | 0,00 | 16,90 | 0,00 |
| f2 + (28) | 7,52 | 0,00 | 7,88 | 0,00 | 6,93 | 0,00 | 11,19 | 0,00 | 11,13 | 0,00 | 9,45 | 0,00 |
| f1 + (30) | 9,38 | 100,00 | 9,18 | 0,00 | 9,18 | 0,00 | 19,02 | 100,00 | 19,01 | 0,00 | 15,90 | 0,00 |
| f2 + (30) | 5,19 | 0,00 | 5,34 | 0,00 | 5,34 | 0,00 | 10,01 | 0,00 | 10,01 | 0,00 | 8,45 | 0,00 |
| f1 + (31) | 9,38 | 100,00 | 9,18 | 0,00 | 9,18 | 0,00 | 19,02 | 100,00 | 19,02 | 0,00 | 15,90 | 0,00 |
| f2 + (31) | 5,19 | 0,00 | 5,34 | 0,00 | 5,34 | 0,00 | 10,01 | 0,00 | 10,01 | 0,00 | 8,45 | 0,00 |
| f3 + (31) | 24,15 | 0,00 | 35,05 | 0,00 | 32,02 | 0,00 | 26,47 | 0,00 | 44,08 | 0,00 | 38,16 | 0,00 |

OF: objective function value; %: relative gap.

**Table 15. Experimental results n = 20 customers (Group C).**

| Customers (n) | 20 | | | | | | | | | | | |
|---|---|---|---|---|---|---|---|---|---|---|---|---|
| Depot | C | | | | | | NC | | | | | |
| Frequency | W (ω = 20) | | M (ω = 30) | | S (ω = 40) | | W (ω = 20) | | M (ω = 30) | | S (ω = 40) | |
| | OF | % | OF | % | OF | % | OF | % | OF | % | OF | % |
| f1 | 33,00 | 100,00 | 61,00 | 100,00 | 73,00 | 100,00 | 49,00 | 100,00 | 63,00 | 100,00 | 91,00 | 100,00 |
| f2 | 21,00 | 0,00 | 45,00 | 73,60 | 72,00 | 82,52 | 30,00 | 33,00 | 46,00 | 70,00 | 75,00 | 77,33 |
| f3 | 67,00 | 0,00 | 121,00 | 9,92 | 134,00 | 2,24 | 89,00 | 22,47 | 153,00 | 28,76 | 178,00 | 26,40 |
| f1 + (28) | 34,00 | 100,00 | 45,99 | 100,00 | 74,99 | 100,00 | 56,00 | 100,00 | 63,90 | 100,00 | 86,99 | 100,00 |
| f2 + (28) | 21,00 | 0,00 | 40,00 | 62,50 | 66,00 | 80,00 | 30,00 | 33,00 | 45,00 | 41,07 | 68,00 | 75,00 |
| f1 + (30) | 33,00 | 100,00 | 55,00 | 100,00 | 72,00 | 100,00 | 49,00 | 100,00 | 69,00 | 100,00 | 84,00 | 100,00 |
| f2 + (30) | 21,00 | 0,00 | 47,00 | 72,19 | 66,00 | 79,04 | 30,00 | 23,33 | 47,00 | 52,20 | 67,00 | 73,13 |
| f1 + (31) | 34,00 | 100,00 | 52,00 | 100,00 | 71,00 | 100,00 | 49,00 | 100,00 | 63,00 | 100,00 | 87,00 | 100,00 |
| f2 + (31) | 21,00 | 42,00 | 42,00 | 69,40 | 58,99 | 75,85 | 30,00 | 47,00 | 46,00 | 58,70 | 70,00 | 67,14 |
| f3 + (31) | 67,00 | 0,00 | 121,00 | 9,09 | 134,00 | 2,98 | 90,00 | 23,33 | 153,00 | 28,10 | 178,00 | 25,84 |

OF: objective function value; %: relative gap.

**Table 16. Experimental results n = 16 customers (Group C).**

| Customers (n) | 16 | | | | | | | | | | | |
|---|---|---|---|---|---|---|---|---|---|---|---|---|
| Depot | C | | | | | | NC | | | | | |
| Frequency | W (ω = 16) | | M (ω = 24) | | S (ω = 32) | | W (ω = 16) | | M (ω = 24) | | S (ω = 32) | |
| | OF | % | OF | % | OF | % | OF | % | OF | % | OF | % |
| f1 | 28,00 | 100,00 | 44,00 | 100,00 | 60,00 | 100,00 | 48,00 | 100,00 | 59,00 | 100,00 | 78,00 | 100,00 |
| f2 | 20,00 | 0,00 | 35,00 | 60,00 | 51,00 | 80,39 | 28,00 | 7,14 | 39,00 | 43,59 | 58,00 | 79,31 |
| f3 | 69,00 | 0,00 | 124,00 | 0,81 | 138,00 | 0,00 | 84,00 | 11,91 | 138,00 | 16,67 | 168,00 | 21,43 |
| f1 + (28) | 34,00 | 100,00 | 44,99 | 100,00 | 64,90 | 100,00 | 57,00 | 100,00 | 62,00 | 100,00 | 73,00 | 100,00 |
| f2 + (28) | 22,00 | 0,00 | 33,00 | 30,00 | 52,00 | 75,00 | 33,00 | 0,00 | 38,00 | 60,52 | 56,00 | 71,40 |
| f1 + (30) | 32,00 | 100,00 | 44,00 | 100,00 | 62,00 | 100,00 | 48,00 | 100,00 | 60,00 | 100,00 | 79,00 | 100,00 |
| f2 + (30) | 20,00 | 0,00 | 33,99 | 58,82 | 51,00 | 70,58 | 28,00 | 0,00 | 36,00 | 22,22 | 55,00 | 70,91 |
| f1 + (31) | 36,00 | 100,00 | 44,00 | 100,00 | 58,00 | 100,00 | 48,00 | 100,00 | 59,00 | 100,00 | 73,00 | 100,00 |
| f2 + (31) | 20,00 | 0,00 | 33,00 | 52,00 | 47,00 | 72,00 | 28,00 | 0,00 | 39,00 | 47,59 | 55,00 | 71,00 |
| f3 + (31) | 69,00 | 0,00 | 124,00 | 0,00 | 138,00 | 0,00 | 84,00 | 0,00 | 136,00 | 13,97 | 168,00 | 19,64 |

OF: objective function value; %: relative gap.

**Table 17. Two-sample mean test p values of comparison of gaps for additional constraints (28), (30), and (31).**

| | Visits | f1 vs f1 + (31) | f2 vs f2 + (31) | f3 vs f3 + (31) | f2 vs f2 + (28) | f2 vs f2 + (30) |
|---|---|---|---|---|---|---|
| Group A | 30 | NA | 0.6853 | 0.4521 | 0.7983 | 0.3385 |
| | 20 | 0.3632 | 0.2013 | 0.2161 | 0.1953 | 0.3212 |
| Group B | 60 | 0.3632 | 0.3937 | 0.703 | 0.1517 | 0.0053 |
| | 40 | NA | 0.0081 | 0.5083 | 0.9096 | 0.3972 |
| | 20 | NA | 0.1228 | 0.2393 | 0.1582 | 0.1767 |
| | 12 | NA | 0.3632 | NA | 0.3632 | 0.3632 |
| | n | | | | | |
| Table 15 | 16 | NA | 0.0854 | 0.1834 | 0.4069 | 0.0511 |
| Table 16 | 20 | NA | 0.6620 | 0.8122 | 0.0797 | 0.0739 |

NA: Not apply due to equal samples means. f1 vs f1 + (28) and f1 vs f1 + (30): NA for the same reason.

t-tests shows that the additional constraints do not have an effect on the quality of the model, since they do not significantly improve the value of the gap.

- Constraint (29), which is devised as a strategy for breaking symmetry, had no effect on the value of the gap reached in any of the analyzed instances, and therefore the replicated results were not included in the Tables 9–14. One possible reason for this is that such a strategy is already covered in the automatic symmetry management techniques currently included in a commercial solver such as Gurobi, which are probably based on sophisticated ideas such as orbital branching [61].

- For $f2$ only, the addition of the set of dis-aggregated constraints (31) achieves a slight decrease in the gap of less than 4% with 40 visits. The same does constraint (28), which prevents a route from containing a single customer, with 60 visits. For $f1$, no instance with $n \geq 16$ gets a gap lower than 100% in one hour, so there is no significant difference between adding valid restrictions or not.

- The addition of valid inequality (30), which establishes that each arc is traversed in a single direction on each route that includes it, had no effect on the value of the gap when $f1$ was used. The addition of such constraint in models with $f2$ only had a positive effect by reducing the value of the gap in instances with more than 40 customers.

- No significant effect of restrictions (30) and (31) can be explained despite the so-called propagation algorithms used in current commercial solvers, in which formulation changes that probably improve performance during the solution stage are detected automatically [58].

Based on the experimental designed shown in Table 7, we performed analysis of variance (ANOVA) to test the significance of the four factors considered (i.e. objective function, centrality, frequency, and number of visits), for each number of visits: 20 and 30 for group A and 12, 20, 40, and 60 for group B. Taking advantages of the previous findings about no significant differences when adding constraint (31), we used those experiments as the second blocked replicate. Detailed ANOVAs are included in S2 Appendix. Among the more relevant findings are:

- With $p$ values <0.001, the size (i.e. either number of visits or $n$) and the objective function are significant in both groups, as it is expected; however, depot location and frequency appeared to be significant in some cases, but explaining the gap only with contributions from less than 1% to 4.8%. Centrality in group B has $p$ value of 0.07.

- The objective function to be solved is the factor with the greatest effect on the value of the gap. As observed in Fig 4, minimizing the total duration of route $f1$ is by far the most difficult objective function to solve. As the size of the problem increases (number of visits or customers), the difference between the effects on the value of the gap of $f1$ versus $f2$ and $f3$ becomes more pronounced. The outstanding performance of $f2$ over $f1$ is confirmed. The gaps of the three objective functions vary almost proportionally; this is, gaps when solving $f2$ are almost 35% greater than when solving $f3$ and 45% less than when solving $f1$.

- For instances with more than 10 customers, the models with $f1$ or $f2$ are more sensitive to the size of the problem (measured in number of customers), than the models with the usual $f3$. With $f1$, only instances with $n \leq 10$ reached gaps lower than 100% as is shown in Tables 13 and 14.

- Gaps are consistently better for semiweekly frequencies. This means that for a given number of visits, a larger number of customers increases the complexity. However, the effect of the frequency type on the gaps is only significant for small instances of Group B, with 12 and

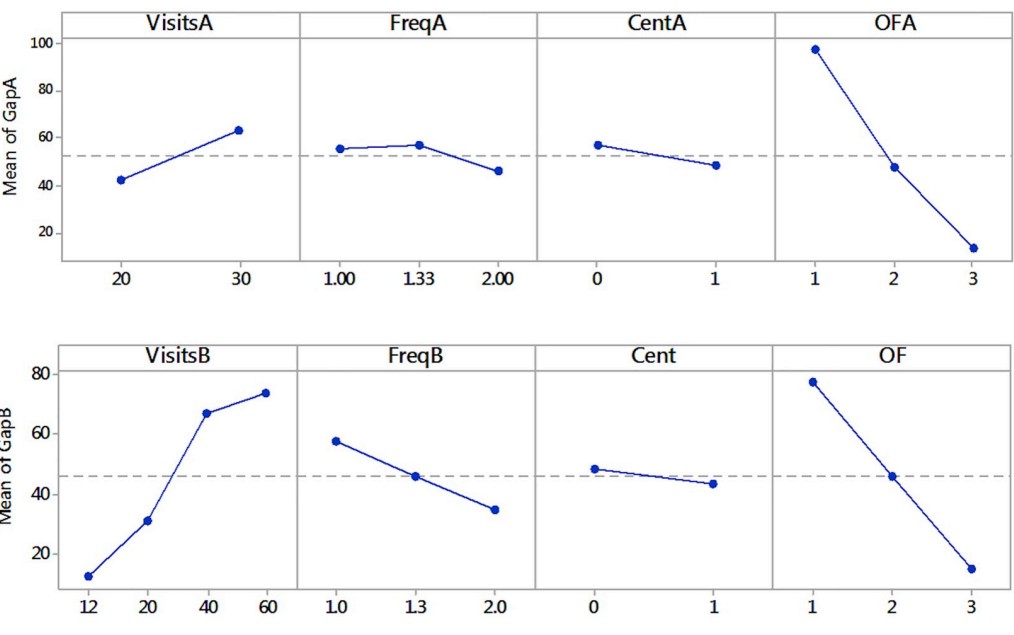

**Fig 4. Main effects plots for gap (fitted means).** Groups A and B, respectively.

20 visits, for which the contribution is close to 10%. For the instances with the highest number of visits, the contribution of this factor is less than 1%. This suggests that the influence of the periodic assignment problem on the complexity of the model is quickly surpassed by the one corresponding to the routing problem.

- In general, the centrality of depot location appears to be not significant. It contributes only 1% to the variability of the gap. Only *f*3 is affected by centrality. When the depot is centered, 20% lower gaps are obtained.

- Significant interactions are identified in both groups. The interaction plots shown in Fig 5 indicate that frequency impacts gaps more when the number of visits is smaller; the effect of the number of visits is accentuated with *f*2; and double visit frequencies impact gaps more when the number of visits is larger and for *f*1. Finally, for *f*3 gaps behave better with centered depot.

Finally, with information from Tables 15 and 16, where the visits increases as the number of customers and the frequency do, the more significant factors are again the size of the

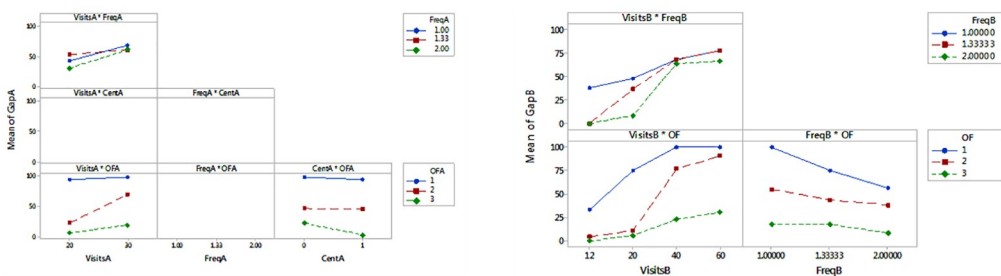

**Fig 5. Interaction plots of significant factors.** Groups A and B, respectively.

problem, measured in frequency and the objective function. Centrality appears now to be significant, suggesting that the gaps are slightly better with centered depot. The frequency affects gaps for $f2$ but not for $f3$.

## Conclusions and further work

This paper presents a MILP model framework for several variants of ConPVRP, and carried out numerical experimentation on one of the models included in the Framework. The framework presented includes models for ConPVRP, ConPVRP-TW, ConTDPVRP, ConTDPVRP-TW. According to our review, the cases of ConTDPVRP and ConTDPVRP-TW had not been previously modeled. For the different models of the framework, two unusual objective functions were formulated and a third objective function of common use was incorporated for comparison purposes.

The ConPVRP-TW, one of the variants included in the framework, was validated through an experimental design where 4 factors were considered: objective function, number of visits, types of frequencies, and depot centrality. The analysis helped identify the significant factors, showed that the models performance is very sensitive to a relatively small sample size increase; and suggests future research directions in this type of problems. It has been demonstrated how the use of the non-conventional objective functions $f1$ and $f2$ lead to models of significantly higher complexity than those in which the conventional function $f3$ is used. One result that is worth noting is that the performance of the models that that minimizes the time in which the last customer is visited $f2$ far exceeds the option of minimizing the maximum duration of a route $f1$. In addition, results showed that the complexity of the problem is better explained by the routing problem than the periodic assignment problem.

Among the potential research lines derived from this work is the improvement of the mathematical formulations for all models considered in the framework, by including adequate cuts that allow finding in less time better lower bounds, with the consequent reduction of relative gaps. Such effort is necessary especially for the objective function $f1$. Finally, it is worthwhile developing a more comprehensive experimental design that incorporates the other types of problems raised in the proposed model framework for ConPVRPs.

## Supporting information

**S1 Appendix. Statistical two-sample difference tests.** It was applied to evaluate the effect of the inclusion of each of the additional constraints (28), (30), and (31) on the gap.
(PDF)

**S2 Appendix. Summary of ANOVA test.** It was applied to evaluate the significance of the four factors considered in the experimental design: objective function, centrality, frequency, and number of visits.
(PDF)

## Acknowledgments

The authors acknowledge supercomputing resources made available by the Centro de Computación Científica Apolo at Universidad EAFIT to conduct the research reported in this scientific product.

## Author Contributions

**Conceptualization:** Maria Gulnara Baldoquin.

**Data curation:** Jairo A. Martinez, Jenny Díaz-Ramírez.

**Formal analysis:** Jenny Díaz-Ramírez.

**Funding acquisition:** Maria Gulnara Baldoquin, Jairo A. Martinez, Jenny Díaz-Ramírez.

**Investigation:** Maria Gulnara Baldoquin, Jairo A. Martinez, Jenny Díaz-Ramírez.

**Methodology:** Maria Gulnara Baldoquin, Jairo A. Martinez.

**Project administration:** Maria Gulnara Baldoquin.

**Resources:** Maria Gulnara Baldoquin, Jairo A. Martinez, Jenny Díaz-Ramírez.

**Software:** Jairo A. Martinez.

**Supervision:** Maria Gulnara Baldoquin.

**Validation:** Jenny Díaz-Ramírez.

**Visualization:** Maria Gulnara Baldoquin.

**Writing – original draft:** Maria Gulnara Baldoquin, Jairo A. Martinez.

**Writing – review & editing:** Jenny Díaz-Ramírez.

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
