## [Decision Letter · Decision Letter 0]

8 Jun 2020

PONE-D-20-06031

A unified model framework for the multi-attribute consistent periodic vehicle routing problem

PLOS ONE

Dear Dr. Díaz-Ramírez,

Thank you for submitting your manuscript to PLOS ONE. After careful consideration, we feel that it has merit but does not fully meet PLOS ONE’s publication criteria as it currently stands. Therefore, we invite you to submit a revised version of the manuscript that addresses the points raised during the review process.

I recommend that it should be revised taking into account the changes requested by Reviewers. I would like to give you a chance to revise your manuscript. To speed the review process, the manuscript will only be reviewed by the Academic Editor in the next round.

We look forward to receiving your revised manuscript.

Kind regards,

Baogui Xin, Ph.D.

Academic Editor

PLOS ONE

Journal Requirements:

2. Please include captions for your Supporting Information files at the end of your manuscript, and update any in-text citations to match accordingly. Please see our Supporting Information guidelines for more information: http://journals.plos.org/plosone/s/supporting-information

Reviewers' comments:

Reviewer's Responses to Questions

**Comments to the Author**

1. Is the manuscript technically sound, and do the data support the conclusions?

Reviewer #1: Partly

Reviewer #2: Partly

2. Has the statistical analysis been performed appropriately and rigorously? 

Reviewer #1: Yes

Reviewer #2: Yes

3. Have the authors made all data underlying the findings in their manuscript fully available?

Reviewer #1: Yes

Reviewer #2: Yes

4. Is the manuscript presented in an intelligible fashion and written in standard English?

Reviewer #1: Yes

Reviewer #2: Yes

5. Review Comments to the Author

Reviewer #1: The manuscript investigates a group of VRP variants, named Multi-Attribute ConPVRP (Consistent Periodic Vehicle Routing Problem) and proceed numerical experiments to explore the impact of different inputs on the results. The research topic is interesting and of great practical significance. However, there are several points in the manuscript that should be better explained and significantly improved in order to be considered for publication.

1. The authors claimed that a unified model framework is proposed. However, from a modeling perspective, it is just a list of multiple models with limited three objective functions. So that I don't think this model is universal.

2. The authors also claimed that the proposed framework includes the model of ConTDPVRP and ConTDPVRP-TW, which have not been previously modeled. I think it is very important and necessary to prove the framework has a significant value in theory and not trivial. Because time dependency and time window problem constraints have individually been well studied, it is not hard to simply mix them in one model.

3. This manuscript selects relative gap as the main performance measure. But from Section 4 I don’t understand what does it mean and how it is calculated. I think it should be defined clearly (preferably expressed in model language).

4. Since a commercial solver (recognized as a black box) is used, I guess the relative gap represents the relative distance between the current feasible solution and a lower bound of a problem. If so, the gap depends on not only the problem input (i.e., frequency of visits per customer, centrality of the depot, and the objective function) but also the solution methods (e.g., branch and bound). How to consider the impact of the solution method on the results?

5. A suggestion. A statement of the problem before introducing the model framework will better can help readers understand the problem better. I hope that the author can first explain some concepts of VRP variants, such as “consistent” and “periodic”.

6. There is some ambiguity in Figure 1. Figure 1 shows that the constraints of TD and TW are independent, but in fact different constraints are selected according to the scenario (ConPVRP, ConTDPVRP, ConPVRP-TW, ConTDPVRP-TW). I suggest Figure 1 can be redrawn to a table as below.

ConPVRP, ConTDPVRP, ConPVRP-TW, ConTDPVRP-TW

Obj1 (Constraint Numbers)

Obj2

Obj3

7. The purpose of the case design needs to be explained more accurately. Specifically, what factor is explored in group A, B, and C?

8. In page 3, last line, the section number is missing.

Reviewer #2: 1. Authors should rewrite the abstract to make it more clearly comprehensible and more complete with necessary items, including aim, method applied, key results obtained, remark, and so on.

2. The contribution of this paper is not clear. Authors should clearly point out the main contribution of the paper, especially its contribution to new knowledge, by comparing it with previous research.

3. Authors should add a section to descript the real problems studied in the manuscript.

6. PLOS authors have the option to publish the peer review history of their article (what does this mean?). If published, this will include your full peer review and any attached files.

Reviewer #1: No

Reviewer #2: No

---

## [Author Response · Author response to Decision Letter 0]

6 Jul 2020

Dear Editor and reviewers

Authors are very thankful for the comments received, which we are sure that helped to improve the quality of the manuscript. Every comment was carefully addressed in the file "Response to Reviewers". Below there is a copy of the text (without any format):

Reviewers' comments

Reviewer #1:

The manuscript investigates a group of VRP variants, named Multi-Attribute ConPVRP (Consistent Periodic Vehicle Routing Problem) and proceed numerical experiments to explore the impact of different inputs on the results. The research topic is interesting and of great practical significance. However, there are several points in the manuscript that should be better explained and significantly improved in order to be considered for publication.

1. The authors claimed that a unified model framework is proposed. However, from a modeling perspective, it is just a list of multiple models with limited three objective functions. So that I don't think this model is universal.

R/ 

We do not just list multiple models. We put together in one single model the variants considered: periodicity, consistency, time-dependency, and time-windows, which have not been simultaneously studied before, according to our literature review. We call it “framework”, because it allows to choose the set of constraints required for a given problem, depending on the variants to be analyzed. Our model connects them. 

On the other hand, we didn’t pretend that the framework is “universal”. Although, this framework is a good starting point to develop and solve practical applications dealing with these variants. Though there are many options for the objective function that can accompany the model constraints, we included three that we considered to be interesting and relevant to practical purposes, two of them poorly explored before. We show in the experimental analysis, how they are of significantly influence in the performance of the models.

To clarify this in the manuscript:

- We added a couple of paragraphs dedicated to the objective functions in the model framework section. We have previously presented them but we didn´t justify them. 

- We added a paragraph describing a practical situation in the introduction section. This follows the reviewer 2’s recommendation, and it helps to give a general context of the possible applications of the framework proposed. 

- We complemented the tables in the literature review, adding the objective functions considered.

The paragraphs to describe and justify the objective functions analyzed in the experimentation section are included before presenting the model, and are as follows: 

The set of constraints shaping all variants considered in this work can be used to optimize the objective function that better fulfill the researcher’s needs. The model framework proposed includes three options of objective functions: 

• (f1) that minimizes the maximum duration of a route.

• (f2) that minimizes the time in which the last customer is visited.

• (f3) that minimizes the total transportation time over the entire planning horizon

(f1) and (f2) are functions of the makespan minimization type. According to Braekers et al. [4], they are not considered standard objective functions although both are based on time or distance. These functions have been used in parcel applications [47], load balancing in home health services [48], manufacturing processing times [49]. Other practical problems where (f2) gains importance is in bus routing, where the maximum travel time of the first student collected in the route wants to be minimized [50, 51]. (f3) has been added in the analysis due that it is one of the standard and most common objective functions explored in VRP. This inclusion will allow future benchmark or experimental comparisons. Examples of VRP studies considering this function are [7, 12, 20–22, 39, 52]

2. The authors also claimed that the proposed framework includes the model of ConTDPVRP and ConTDPVRP-TW, which have not been previously modeled. I think it is very important and necessary to prove the framework has a significant value in theory and not trivial. Because time dependency and time window problem constraints have individually been well studied, it is not hard to simply mix them in one model.

R/ 

We agree that the time-dependency and time-window variants of VRP are very well studied problems. They usually use typical objective functions such as minimizing the total distance traveled and minimizing the number of vehicles, as shown in Table 1. The literature review section also reveals that these problems have been poorly addressed when using non-standard objective functions such as minimizing the makespan. As we said in the previous reply, our framework responds to these gaps. It not only puts these two variants together, but adds the complexity due to other variants that have been also studied separately: periodicity, and consistency, and involves not only the standard objective function of minimizing total traveled time but two non-conventional objective functions of practical relevance, whose effects on the model solution are subsequently analyzed. 

Following the reviewer’s suggestion (query #5), we revised the formal definition of the concepts of consistency and periodicity in the introduction section. Our numerical experimentation focuses on these two variants since time-dependency and time-windows have been more studied. The experimental setup considers the frequency as a design factor, which is the one that determines the number of visits to each customer during the planning horizon. The consistency aspect is considered throughout the model.

In addition, in the reply to Reviewer 2’ query #2, we added the revised paragraphs that outline the main contributions of the paper.

3. This manuscript selects relative gap as the main performance measure. But from Section 4 I do not understand what does it mean and how it is calculated. I think it should be defined clearly (preferably expressed in model language).

R/ We followed this recommendation. 

We added a sub-section in the Experimental setup section dedicated to the performance measures. They were previously mentioned and described in the Results section, so we moved two paragraphs up to this new section, and complemented it with one paragraph dedicated to the definition of the “gap” measure. 

The entire new section included in the revised version of the manuscript is as follows:

Performance measures

In this work, we solved the different instances that we have just described using the general-purpose MILP solver Gurobi-8.1.1. The modeling language used was Pyomo. The experiments were run in a computer with 8 CPUs Intel®Xeon®E5-2670 2.60GHz, and the operating system Linux Rocks 6.2. Under the same parameter tuning of the solver, modeling performance is assessed with two values: the objective function value (OF) and the relative gap (%). Table 7 shows the values used for some of the most important parameters that can influence the solution process of a MILP model [48]. 

The branch-and-bound (B&B) algorithms used by the solver to solve MIP problems keep the best integer solution found along with its objective function value z ^ which is called the incumbent solution. If the problem is a minimization one, z ^ is an upper bound for the optimal solution of the original MILP formulation. Further details of the B&B algorithms can be found in [55]. At any time during the B&B search, there is also a lower valid bound, called the best bound z*, which is obtained by taking the minimum of the optimal values of the objective function on the leaf nodes. The absolute gap is the difference between the bounds, i.e. gap = z ^ –z*. The relative gap is obtained by dividing the absolute gap by the best lower bound, that is rel_gap = gap/z*. When the gap is less than a small value �, the incumbent solution is returned as the optimal for the original problem [54, 56].

The choice of the relative gap as the main performance measure is consistent with Klotz & Newman [57], who showed that through careful formulation and algorithmic parameter tuning, the optimizer performance can be improved in terms of the optimality gaps (%). In addition, recognizing that commercial solvers are largely a black box, studying the effect of certain model constraints, remains a valid research question [58]. On other hand, the OF serves to compare variants of the same model, and is the typical performance measure of optimization models (for example, [59, 60]). Results reported are the values obtained after a fixed maximum computation time of one hour. Time selection obeyed to a preliminary experimentation with 10-hour runs, where it could be observed that the best lower bounds where achieved in this period, and following [57], good lower bounds better reflect the difficulty of a model solution than other aspects of the solution process.

4. Since a commercial solver (recognized as a black box) is used, I guess the relative gap represents the relative distance between the current feasible solution and a lower bound of a problem. If so, the gap depends on not only the problem input (i.e., frequency of visits per customer, centrality of the depot, and the objective function) but also the solution methods (e.g., branch and bound). How to consider the impact of the solution method on the results?

R/ We agree.

We have just mentioned in the previous reply that we added a formal description of relative gap. We agree that the solution methods the solver can use internally are varied, and that they can introduce a noise when the performance is measured with gaps. To minimize the impact, we made sure the solver did not “decide” in some of the solver options, but we fixed them. We added the Table 6 that includes a description of some of the most relevant solver option values used. This table is included in the new section “Performance measures” presented in previous reply.

5. A suggestion. A statement of the problem before introducing the model framework will better can help readers understand the problem better. I hope that the author can first explain some concepts of VRP variants, such as “consistent” and “periodic”.

R/ Both terms were already referred to in the introduction when we mentioned the Periodic VRP and the Consistent VRP. Following the suggestion, these definitions were revised as follows: 

Periodic (revised): 

“...the PVRP looks for building a plan of optimal routes for the entire planning horizon (i.e. more than one day), knowing in advance the frequency of visits demanded by each customer. It involves deciding the pattern of visits for each customer, the selection of the vehicles that visit each customer, and the visit order.” 

Consistent (revised):

“In [9] a consistent VRP (ConVRP) considers that the same driver visits the same customers throughout the planning horizon, at roughly the same time on each day that these customers are visited. In [10] a generalized ConVRP is considered, where each customer is visited by a limited number of drivers and the variation in the arrival times is penalized in the objective function. A collection of vehicle routing problems in which consistency considerations are relevant are described in [11].” 

In addition, the introduction statement in the model framework section was revised. 

The model framework proposed in this work considers the following variants of the ConPVRP: ConPVRP-TW, ConTDPVRP and ConTDPVRP-TW. First, the structure of the model framework is schemed in Fig 1. Next, the objective functions selected are described and justified, and finally, the model framework is presented in detail.

Next to previous text and before presenting the model, the revised paragraphs about the objective functions were included. They were previously shown in query #1’s reply.

6. There is some ambiguity in Figure 1. Figure 1 shows that the constraints of TD and TW are independent, but in fact different constraints are selected according to the scenario (ConPVRP, ConTDPVRP, ConPVRP-TW, ConTDPVRP-TW). I suggest Figure 1 can be redrawn to a table as below.

R/ We built a table representing the same idea as Figure 1, as shown below. However, the authors still believe that the figure offers an easier reading of the combination of variants included in the framework, rather than the table. So, we kept and revised the figure and the text that refers to it. The table was also included as supplementary information. We present below both options. 

Optional Table to replace Figure 1:

Revised Figure 1 in manuscript:

Fig. 1. Model Framework Scheme

ConPVRP: Consistent periodic VRP, TD: time-dependence, TW: time-windows, AV: additional valid equations

The final description of the figure, which we believe gives a much clearer way of reading it is as follows:

Fig 1 schematizes the variants that are analyzed, and distinguishes the specific constraints of each variant from the core constraints of the ConPVRP, by indicating the number of the equations that constitute each one. The core VRP constraints reflecting consistency and periodicity are included in the middle box ConPRVP. Constraints in the box TD are exclusively used if the model considers time-dependence. The same works for the time-windows variant in the box TW. The additional valid constraints that were revised in the numerical experimentation are also optional, and presented in the box AV.

The set of constraints shaping all variants considered in this work can be used to optimize the function that better fulfil the researcher’s needs. The objective functions equations included in the framework, and the constraints needed to connect them with the rest of the model, are indicated in their respective box in Fig 1.

6. The purpose of the case design needs to be explained more accurately. Specifically, what factor is explored in group A, B, and C?

R/ Descriptions of the groups were revised and complemented to make them clearer. The final description is as follows:

Three groups were generated with a total of eight data sets in square areas chosen arbitrarily. The customer locations were randomly assigned following operational circumstances consistent to the real problem addressed. For example, group A instances seek to include situations where distribution is over suburban areas, comprising a relatively larger area and a low density of customers. In contrast, group B instances are intended to reflect cases of "last mile distribution" in which zoning involves small distribution areas and varying customer density over a wider range. Two data sets conform group C that use group A setting to test the effect of the number of visits ω when the number of customers n is fixed.

7. In page 3, last line, the section number is missing.

R/ Though in our submitted version it was numbered, this new version was revised such that all lines were numbered, including the abstract.

 

Reviewer #2: 

1. Authors should rewrite the abstract to make it more clearly comprehensible and more complete with necessary items, including aim, method applied, key results obtained, remark, and so on.

R/ We followed the recommendation. The abstract was completely revised.

New Abstract:

Modeling real-life transportation problems usually require the simultaneous incorporation of different variants of the classical vehicle routing problem (VRP). The periodic VRP (PVRP) is a classical extension in which routes are determined for a planning period of several days and each customer has an associated set of allowable visit schedules. This work proposes a unified model framework for PVRP that consists of multiple attributes or variants not previously addressed simultaneously, such as multi-depot, time-windows, time-dependence, and consistency -which guarantees the visits to customer by the same vehicle-, together with three objective functions that respond to the needs of practical problems. The numerical experimentation is focused on the effects of three factors: frequency, depot centrality, and the objective function on the performance of a general–purpose MILP solver, through the analysis of the achieved relative gaps. Results show higher sensitivity to the objective functions and to the problem sizes.

2. The contribution of this paper is not clear. Authors should clearly point out the main contribution of the paper, especially its contribution to new knowledge, by comparing it with previous research.

R/ The abstract and introduction sections outline the main contributions of the paper. These were revised to make them clearer. In previous reply we presented the new abstract, where we describe the novelty, the variants included in the framework, the experimental design, and insights of the numerical experimentation results. 

The introduction was also revised. These are the modified paragraphs:

This paper makes two contributions to literature, being the first one of modeling-type, and the second one a numerical-type contribution; as follows:

- A unified model framework for the multi-attribute ConPVRP inspired by real problems. The framework includes variants, and their relationships, not considered simultaneously before. In addition, it includes the analysis of three objective functions, two of them uncommonly discussed but inspired by real problems.

- The experimentation design includes the simultaneous analysis of three relevant factors to these types of problems: frequency, depot centrality, and the objective functions. We provide experimental evidence of how the two non-conventional objective functions are harder problems to solve, and that some active constraints found in the literature actually do not improve the performance of the model.

3. Authors should add a section to descript the real problems studied in the manuscript.

 R/ We followed the recommendation. A paragraph in the introduction section was added:

A general real-world context that inspires the development of the framework for ConPVRP is as follows: The company has a set of sale points (or customers) that must be visited at a frequency determined by its sales volume (or demand) by one of the trucks of the company fleet. There are 4 types of visit frequency: weekly (the same day each week, for example every Tuesday), biweekly (two visits per week, e.g. Monday and Thursday or Tuesday and Friday), bimonthly (2 times a month, e.g. in the first and third weeks or the second and fourth weeks, but always on the same day of the week), and monthly. Trucks must start and finish their journey at the central depot. Each customer must always be visited by the same truck. Several customers can be visited on the same day by the same truck. The trucks are available from Monday to Friday. Though the trucks have a limited capacity, it is assumed that the total available travel time is the dominant constraint [2]. This is why f1 and f2 become relevant objective functions.

---

## [Decision Letter · Decision Letter 1]

20 Jul 2020

A unified model framework for the multi-attribute consistent periodic vehicle routing problem

PONE-D-20-06031R1

Dear Dr. Díaz-Ramírez,

We’re pleased to inform you that your manuscript has been judged scientifically suitable for publication and will be formally accepted for publication once it meets all outstanding technical requirements.

Kind regards,

Baogui Xin, Ph.D.

Academic Editor

PLOS ONE

Additional Editor Comments (optional):

Reviewers' comments:

Reviewer's Responses to Questions

**Comments to the Author**

1. If the authors have adequately addressed your comments raised in a previous round of review and you feel that this manuscript is now acceptable for publication, you may indicate that here to bypass the “Comments to the Author” section, enter your conflict of interest statement in the “Confidential to Editor” section, and submit your "Accept" recommendation.

Reviewer #1: All comments have been addressed

Reviewer #2: All comments have been addressed

2. Is the manuscript technically sound, and do the data support the conclusions?

Reviewer #1: Yes

Reviewer #2: Yes

3. Has the statistical analysis been performed appropriately and rigorously? 

Reviewer #1: Yes

Reviewer #2: Yes

4. Have the authors made all data underlying the findings in their manuscript fully available?

Reviewer #1: Yes

Reviewer #2: Yes

5. Is the manuscript presented in an intelligible fashion and written in standard English?

Reviewer #1: Yes

Reviewer #2: Yes

6. Review Comments to the Author

Reviewer #1: I'm pleased to read the new version with most problems addressed. But I still suggest the table presentation of Figure 1.

Reviewer #2: (No Response)

7. PLOS authors have the option to publish the peer review history of their article (what does this mean?). If published, this will include your full peer review and any attached files.

Reviewer #1: No

Reviewer #2: No

---

## [Editor Report · Acceptance letter]

23 Jul 2020

PONE-D-20-06031R1 

A unified model framework for the multi-attribute consistent periodic vehicle routing problem 

Dear Dr. Díaz-Ramírez:

I'm pleased to inform you that your manuscript has been deemed suitable for publication in PLOS ONE. Congratulations! Your manuscript is now with our production department. 

Kind regards, 

on behalf of

Professor Baogui Xin 

Academic Editor

PLOS ONE